# ClimateAgent: Multi-Agent Orchestration for Complex Climate Data Science Workflows

**Chenyue Li**[*]                                                                          *clieh@connect.ust.hk*
*The Hong Kong University of Science and Technology*

**Hyeonjae Kim**[*]                                                                        *hkimar@connect.ust.hk*
*The Hong Kong University of Science and Technology*

**Wen Deng**                                                                               *wdengan@connect.ust.hk*
*The Hong Kong University of Science and Technology*

**Mengxi Jin**                                                                             *mjinaf@connect.ust.hk*
*The Hong Kong University of Science and Technology*

**Wen Huang**                                                                              *whuangbp@connect.ust.hk*
*The Hong Kong University of Science and Technology*

**Mengqian Lu**                                                                            *mengqian.lu@ust.hk*
*The Hong Kong University of Science and Technology*

**Binhang Yuan**[†]                                                                        *biyuan@ust.hk*
*The Hong Kong University of Science and Technology*

**Reviewed on OpenReview:** *https://openreview.net/forum?id=XLWvXNumGa*

## Abstract

Climate science demands automated workflows to transform comprehensive questions into data-driven statements across massive, heterogeneous datasets. However, generic LLM agents and static scripting pipelines lack climate-specific context and flexibility, thus, perform poorly in practice. We present CLIMATEAGENT, an autonomous *multi-agent* framework that orchestrates end-to-end climate data analytic workflows. CLIMATEAGENT decomposes user questions into executable sub-tasks coordinated by an ORCHESTRATE-AGENT and a PLAN-AGENT; acquires data via specialized DATA-AGENTS that dynamically introspect APIs to synthesize robust download scripts; and completes analysis and reporting with a CODING-AGENT that generates Python code, visualizations, and a final report with a built-in self-correction loop. To enable systematic evaluation, we introduce CLIMATE-AGENT-BENCH-85, a benchmark of 85 real-world tasks spanning atmospheric rivers, drought, extreme precipitation, heat waves, sea surface temperature, and tropical cyclones. On CLIMATE-AGENT-BENCH-85, CLIMATEAGENT achieves 100% task completion and a report quality score of 8.32, outperforming GITHUB-COPILOT (6.27) and a GPT-5 baseline (3.26). These results demonstrate that our multi-agent orchestration with dynamic API awareness and self-correcting execution substantially advances reliable, end-to-end automation for climate science analytic tasks. The source code of CLIMATEAGENT is available at [https://github.com/Relaxed-System-Lab/ClimateAgent].

---

[*]Equal contribution.
[†]Corresponding author.

# 1 Introduction

The rapid pace of climate change and the growing severity of its impacts have created an urgent need to advance data-centric climate science, which could deliver timely insights for adaptation and policy (Deelman et al., 2009; Overpeck et al., 2011; King et al., 2009). Automating the workflows in climate science can translate comprehensive analytical questions into executable pipelines, enabling rapid, reproducible analysis of complex environmental phenomena and supporting extreme-event forecasting, impact assessment, and adaptation planning. On the other hand, such analytic workflows need to process special climate datasets (e.g., datasets from Copernicus climate data store (CDS) and the European centre for medium-range weather forecasts (ECMWF)) with high volume, heterogeneity, and complexity, where intelligent automation has become essential for processing and integrating these datasets at scale (Hersbach et al., 2020; Benestad et al., 2017; Buizza et al., 2018). In this paper, we aim to explore *how AI agents can perform complex climate-science analytical tasks through carefully designed agentic orchestration.*

Building an AI-driven climate agent is a compelling and crucial effort since the stakes of climate change demand faster and more intelligent ways to derive insights from the complex data. The accelerating pace of global change and the severity of its impacts suggest that climate scientists and policymakers urgently need timely, data-driven information for mitigation and adaptation decisions (Overpeck et al., 2011; King et al., 2009). At the same time, climate science workflows need to process massive and diverse datasets — from multi-model simulations to satellite and in-situ observations — and turning this data deluge into useful information has become a bottleneck for discovery and decision-making. A large language model (LLM) based AI agent can automate complex analytical workflows to address this bottleneck, dramatically accelerating analysis that would otherwise take a team of experts days or weeks. By translating high-level scientific questions into executable pipelines, such an agentic paradigm could enable rapid, reproducible analysis of complicated climate phenomena like extreme events, climate impacts, and future scenarios (Deelman et al., 2009; Overpeck et al., 2011). In essence, an AI climate agent can serve as a productive research assistant that integrates data from various sources on-the-fly and explores many potential hypotheses iteratively, which would augment human scientists' abilities, i.e., allowing them to focus on interpretation and strategy, and accelerate the cycle of the analytic workflow (King et al., 2009; Reichstein et al., 2019). By empowering more interactive and comprehensive data exploration, such an agent could usher in a new paradigm for climate science, where insights emerge at the pace of computational power rather than human labor.

On the other hand, developing a robust climate agent is challenging due to several inherent complexities of this domain, where data volume and heterogeneity are primary obstacles: modern climate datasets are enormous in size and varied in format. For example, a single state-of-the-art reanalysis (ERA5) encompasses petabytes of multidimensional data (Hersbach et al., 2020), and observational records, climate model outputs, and remote sensing products each come with different resolutions, units, and conventions (Benestad et al., 2017). A simple, hand-crafted pipeline or off-the-shelf use of LLMs (e.g., standard zero-shot LLM approaches) will quickly break down when faced with such diversity and complexity. Note that many climate workflows require on-the-fly decision making and expert knowledge — for instance, selecting appropriate data sources, applying bias corrections, or choosing relevant statistical tests to determine significance. A rigid, hand-crafted solution cannot easily accommodate these nuanced choices — attempts to force flexibility into static pipelines often lead to brittle, error-prone processes, highlighting the need for more intelligent, adaptive automation (Buizza et al., 2018).

In this paper, we view automating complex climate scientific workflows as a specialized form of planning and data-processing code generation. Recent LLM-based agentic systems show promising performance, but those applied to scientific domains still face critical shortcomings (Austin et al., 2021). Most current agents use generic, domain-agnostic LLMs that miss the specialized requirements of climate science (Wang et al., 2024; Yao et al., 2022). Concretely, a general LLM agent could be unaware of climate data access APIs, data formats, and valid parameter choices, while static scripts and libraries lack the flexibility to handle new scientific questions or evolving data sources. These limitations result in high error rates for LLM-generated code and a steep learning curve for non-expert users. More importantly, generic approaches cannot support the iterative, hypothesis-driven nature of climate study — they fail to autonomously handle multi-step inquiries where initial results could spur new sub-questions — leaving a substantial gap in achieving truly end-to-end, autonomous climate analysis without requiring human intervention.

To address these gaps, we introduce CLIMATEAGENT, an autonomous multi-agent framework specially designed for climate science workflows. Our approach employs a multi-agent orchestration strategy: a comprehensive climate query is first decomposed into a structured sequence of sub-tasks, where each sub-task is executed by a specialized agent. This modular design injects domain-specific knowledge at every step and dynamically adapts as the workflow progresses. Additionally, CLIMATEAGENT is also equipped with robust error handling and self-correction, enabling it to detect failures, recover, and adjust plans without human intervention in the loop. Concretely, we make the following contributions:

**Contribution 1.** We propose a multi-agent orchestration paradigm to support complex climate data science workflows, which comprises the following key agents:

- **Agents for planning and orchestration**: We introduce an ORCHESTRATE-AGENT for workflow management and a PLAN-AGENT for task decomposition. Concretely, the ORCHESTRATE-AGENT manages experiment directories and persistent context, while the PLAN-AGENT interprets the user request, formulates a detailed execution plan, breaks down the high-level goal into discrete subtasks, and delegates each to the appropriate specialist agent. Together, these two agents provide top-level oversight, adjusting the plan as needed and ensuring the overall workflow stays on track, including handling runtime issues or re-planning when necessary.

- **Agents for climate data acquisition**: We implement a set of DATA-AGENTs each tailored to a specific data source and handles data retrieval by dynamically introspecting its target API — e.g., fetching the latest valid parameters or dataset metadata at runtime — and then generating robust download scripts. This capability allows users to adapt to evolving datasets and API utilization, preventing errors such as invalid parameter use or format mismatches during data acquisition.

- **Agents for programming and visualization**: We include a set of CODING-AGENTs that generate Python code for data analysis and generate final reports, complete with text summaries and visualizations. The CODING-AGENTs implement a self-correction loop to debug code based on execution feedback.

Together, these agents form a cohesive system that can autonomously manage the entire climate data science life-cycle from data acquisition to final analysis, making sophisticated climate science accessible and efficient.

**Contribution 2.** To systematically evaluate the performance over the climate data science tasks, we introduce CLIMATE-AGENT-BENCH-85, a benchmark of 85 real-world climate workflow tasks spanning six domains, including atmospheric rivers (AR), drought (DR), extreme precipitation (EP), heat waves (HW), sea surface temperature (SST), and tropical cyclones (TC). Each task is specified in natural language with an explicit scientific objective, required datasets, mandatory external tools (e.g., `TempestExtremes`), and strict output contracts (filenames and formats), driving multi-step pipelines from data acquisition through processing, analysis, and visualization. For reproducibility, every task includes a curated reference solution with a validated Python code base and a human-readable report with figures, and tasks are stratified by difficulty — *easy* (single-source), *medium* (multi-source), and *hard* (external-tool integration with dynamic parameters) — to support controlled comparisons across complexity levels.

**Contribution 3.** We comprehensively evaluate the proposed multi-agent framework, CLIMATEAGENT, on CLIMATE-AGENT-BENCH-85, where the experimental results indicate that CLIMATEAGENT demonstrates the ability to autonomously plan and execute the entire workflow. To be specific, our system achieves a 100% success rate in generating the report across the benchmark, with an overall report quality score of 8.32, compared to 6.27 for GITHUB-COPILOT and 3.26 for the GPT-5 baseline. We believe CLIMATEAGENT significantly reduces the manual effort and specialized expertise required, demonstrating a powerful new paradigm for AI-driven climate data science discovery and advancing the state-of-the-art in automated climate research.

## 2 Related Work

**LLM-Based Agentic Systems and Scientific Automation.** Large language models such as GPT-3 (Brown et al., 2020), Llama 2 (Touvron et al., 2023), and PaLM (Chowdhery et al., 2023) have become the backbone for agentic systems that interpret user instructions, plan tasks, and interact with external APIs

(Yao et al., 2022; Shinn et al., 2023). Frameworks like AutoGPT (Richards, 2023), LangChain (Chase, 2023), and CrewAI (CrewAI, 2023) enable multi-step workflows and collaborative agents, while recent advances have extended these systems with memory, tool usage, and inter-agent collaboration capabilities (Chen et al., 2024; Park et al., 2023). In scientific domains, systems such as ChemCrow (Bran et al., 2024) and SciFact (Wadden et al., 2020) demonstrate automated protocol generation and fact checking. However, these approaches are generally evaluated in synthetic domains or require extensive human oversight. What remains missing is a system that combines domain-specific knowledge integration, persistent workflow state management, and robust error recovery for end-to-end scientific analysis with live, evolving data sources.

**Climate Science Workflow Automation.** Workflow automation in climate science has evolved from generic orchestrators like Kepler (Altintas et al., 2004) and Apache Airflow to domain-adapted engines such as Cylc (Oliver et al., 2018) and ESMValTool (Righi et al., 2020). Programmatic access to climate datasets is enabled by libraries like cdsapi, ecmwf-api-client (Copernicus Climate Change Service (C3S), 2019; European Centre for Medium-Range Weather Forecasts (ECMWF), 2022), and processing interfaces like CDO and xarray (Hoyer & Hamman, 2017). Recent specialized systems have begun targeting autonomous climate analysis: ClimSim-Online (Sridhar et al., 2023) explored goal-driven automation for climate impact modeling, while TorchClim (Fuchs et al., 2024) and EarthML (Project, 2020) focus on deep learning-powered workflows. However, existing systems either require manual intervention at each workflow stage or lack the robustness to handle the dynamic nature of climate APIs and heterogeneous data sources, preventing truly autonomous end-to-end analysis.

**Autonomous Research and Multi-Agent Planning.** Advanced LLM architectures incorporating meta-prompting (Zhang et al., 2025c), chain-of-thought reasoning (Wei et al., 2022), and self-correction feedback loops (Kamoi et al., 2024) have improved task decomposition and reliability in autonomous research systems. Prompt-based approaches like PromptCast (Xue & Salim, 2022) and LLMTime (Gruver et al., 2023) demonstrate zero-shot capabilities in time series analysis, while automated debugging agents (Gao et al., 2024) enhance code reliability. However, the gap between high-level planning capabilities and execution-level robustness remains unaddressed: few systems successfully translate broad user goals into complex multi-agent plans with error-recoverable execution for scientific APIs, particularly in climate science.

To address these gaps, i.e., domain-agnostic agents, brittle workflow execution, and lack of comprehensive evaluation benchmarks, we introduce CLIMATEAGENT, a specialized multi-agent framework that embeds climate expertise at every workflow stage while maintaining the flexibility to recover from inevitable failures.

## 3 ClimateAgent

Climate science workflows present a fundamental challenge for automation: researchers must coordinate multiple interdependent tasks across heterogeneous data sources, each requiring specialized domain knowledge. A typical workflow begins with identifying and acquiring data through diverse APIs — reanalysis products, forecast models, observational datasets — then proceeds through multi-dimensional data processing, statistical analysis, and finally report generation. Each phase demands expertise in domain-specific conventions: climate data APIs impose unique constraints on spatial and temporal queries, scientific computing libraries require precise parameter configurations, and analysis must respect the statistical properties of geophysical data. Existing LLM-based approaches cannot handle this complexity: single-agent systems lack the specialized knowledge required at each stage, while monolithic code generation cannot recover from API violations, parameter errors, or dependency failures.

We observe that the structure of climate workflows naturally suggests a solution. Rather than attempting monolithic generation, we can decompose complex analyses into specialized subtasks — mirroring how climate researchers organize their own work into data acquisition, processing, analysis, and reporting phases. By assigning each subtask to an expert agent with targeted domain knowledge, and coordinating execution through shared context that accumulates artifacts and enables iterative refinement, we leverage LLMs' code generation capabilities while building in the error recovery mechanisms that long scientific workflows require.

We present CLIMATEAGENT, an autonomous multi-agent framework that executes end-to-end climate analyses without human intervention. CLIMATEAGENT decomposes workflows into specialized agents that collabo-

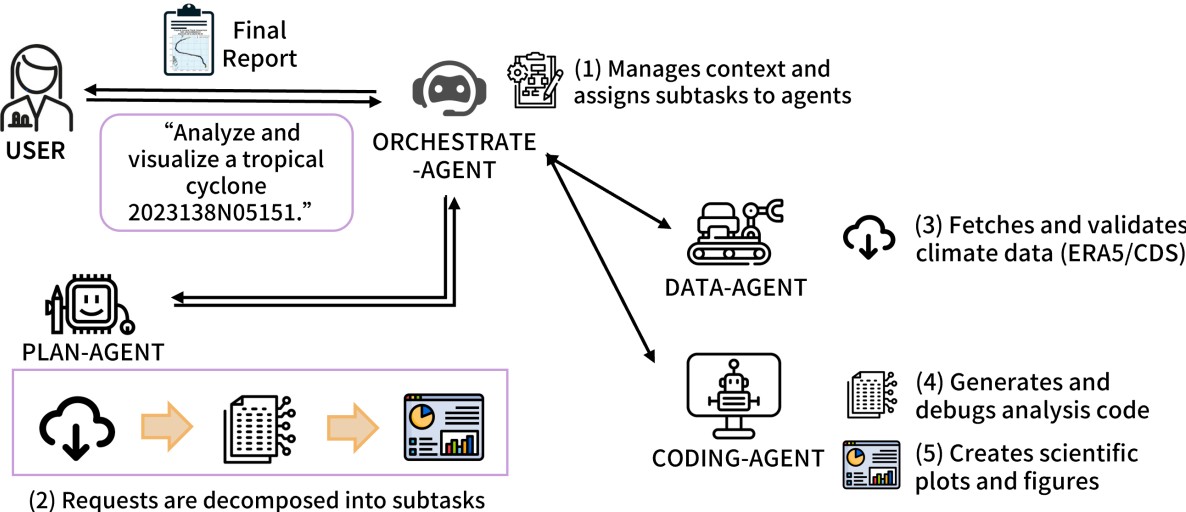

Figure 1: Overview of the CLIMATEAGENT system architecture. The workflow illustrates how user queries are decomposed and processed by specialized agents, with robust error recovery and context management, to produce comprehensive climate science reports.

rate through shared context and adaptive feedback loops. The ORCHESTRATE-AGENT coordinates execution while the PLAN-AGENT decomposes queries into executable subtasks. DATA-AGENTs introspect climate APIs to generate validated download scripts, and CODING-AGENTs produce analysis code with built-in debugging and self-correction. To achieve truly autonomous scientific workflows, CLIMATEAGENT realizes three core capabilities that address the fundamental requirements of climate analysis automation:

- **Coordinated Task Planning.** The PLAN-AGENT decomposes comprehensive climate queries into structured subtasks and delegates each to specialized agents with targeted domain expertise, forming a cooperative division of labor that improves the completeness and quality of the resulting analysis and reports.

- **Contextual Coordination.** Agents operate on a shared, persistent context that enables cross-step communication and propagation of intermediate results, dynamically adapting as the workflow progresses and maintaining methodological consistency across multi-stage analyses.

- **Adaptive Self-Correction.** CLIMATEAGENT incorporates built-in error detection and recovery mechanisms that allow agents to diagnose failures, revise plans, and adapt without human intervention, ensuring robustness under evolving data and API conditions.

In this section, we first formalize climate report generation as sequential state transformation (§3.1), then describe how our three-layer architecture achieves **Coordinated Task Planning** (§3.2), how persistent context enables **Contextual Coordination** (§3.3), and how complementary error recovery strategies provide **Adaptive Self-Correction** (§3.4). Finally, we synthesize these mechanisms into a unified orchestration algorithm (§3.5). We also provided detailed CLIMATEAGENT implementation & reproducibility in Appendix B.

## 3.1 Problem Formulation

We formalize climate workflow execution as a sequential state transformation. Given a task $T$, the system produces a scientific report $R$ as the final output of a multi-stage analytical workflow through coordinated multi-agent operations that maintain a persistent workflow context $\mathcal{C}_i$ accumulating all generated artifacts:

$$\mathcal{C}_i = \{\text{task} : T, \text{plan} : P, \text{code} : \{c_j\}_{j=1}^i, \text{data} : \{d_j\}_{j=1}^i, \text{results} : \{r_j\}_{j=1}^i, \text{logs} : \{l_j\}_{j=1}^i\}$$

The PLAN-AGENT decomposes $T$ into an ordered sequence of subtasks $P = [s_1, s_2, \ldots, s_n]$, each specifying the action, required data, and target agent. Specialized agents $A_k$ execute subtasks via the transition:

$$\mathcal{C}_i = \text{Execute}(\mathcal{C}_{i-1}, s_i, A_k)$$

This design achieves *workflow coherence through context accumulation*: each agent operates on the complete history of prior decisions and artifacts. The context serves as (1) an inter-agent communication protocol, (2) a checkpoint for workflow resumption, and (3) a provenance record for reproducibility.

## 3.2 Multi-Agents for Coordinated Task Planning

To realize the coordinated task planning capability, CLIMATEAGENT employs a three-layer hierarchical architecture (see Figure 1) that mirrors how climate scientists organize collaborative investigations. Specialized agents at each layer encode distinct domain expertise: the PLAN-AGENT and ORCHESTRATE-AGENT coordinate workflow execution, DATA-AGENTs handle data acquisition, and CODING-AGENTs perform analysis and synthesis.

**Agents for Planning and Orchestration**. ORCHESTRATE-AGENT manages workflow execution (Step 1 in Figure 1) by creating timestamped experiment directories, persisting context, and coordinating agent invocation. PLAN-AGENT decomposes queries into executable subtasks using climate-domain reasoning: it recognizes standard analysis patterns (climatology → anomalies → extremes → report), dataset dependencies (reanalysis vs. forecasts), and temporal constraints (initialization dates, lead times, aggregation windows).

**Agents for Climate Data Acquisition**. DATA-AGENTs interface with heterogeneous climate data sources (Step 2) through specialized implementations. The CDS variant uses `cdsapi` to access the Copernicus Climate Data Store, validating variables, time ranges, and spatial domains. The ECMWF variant uses `ecmwf-api-client` to retrieve S2S forecasts, encoding knowledge of model origins (ECMWF, NCEP, JMA), parameter types (pressure-level, surface, daily-averaged), and forecast conventions.

**Agents for Programming and Visualization**. CODING-AGENTs generate Python code (Steps 3-5) for data processing and report visualization. They incorporate expertise in climate libraries (xarray, cartopy, cf-python), domain-appropriate statistical methods (anomaly calculations, running means, extreme value statistics), and scientific computing practices (vectorized operations, memory management, unit handling).

This three-layer architecture separates concerns that require distinct expertise: data acquisition demands API knowledge, processing requires computational skills, and reporting requires synthesis ability. Encoding domain expertise at each layer achieves modularity and reliability.

## 3.3 Contextual Coordination

Building upon the hierarchical architecture in §3.2, CLIMATEAGENT achieves the second core capability, i.e., contextual coordination, by maintaining a persistent and interpretable workflow memory shared across all agents. This shared context records plans, data artifacts, and execution outputs, enabling continuity, communication, and reproducibility throughout the workflow. Formally, the persistent workflow context $\mathcal{C}_i$ serves as the central mechanism that connects agents, supports error recovery, and preserves the system state. Context functions as both a communication protocol and a state management system.

**Context Evolution.** In this procedure, each agent receives $\mathcal{C}_{i-1}$ and produces $\mathcal{C}_i$ by appending new artifacts (scripts, outputs). This monotonic accumulation preserves all information across agent transitions. For example, CODING-AGENT accesses original data files, processed results, and prior code when generating visualizations.

**Context-Driven Code Generation.** The agents should then leverage context to generate coherent code. Concretely, CODING-AGENT examines $\mathcal{C}_{i-1}$ to discover available files, understand data structure, and maintain consistent naming conventions. This prevents common errors: hard-coded paths, incorrect variable assumptions, and redundant computations.

**Serialization and Persistence.** We serialize context to JSON after each subtask, enabling: (1) *workflow resumption* from the last completed subtask after interruption; (2) *reproducibility* by preserving complete

decision history; (3) *debugging* without re-executing earlier stages. This persistent state transforms independent agents into a coherent, traceable workflow engine.

### 3.4 Adaptive Self-Correction

Building on the contextual coordination mechanisms in §3.3, CLIMATEAGENT realizes the third core capability, i.e., adaptive self-correction, which provides robustness and adaptability under real-world scientific constraints. Note that climate workflows could face unexpected execution failures: APIs impose inconsistent parameter rules, data availability fluctuates, and generated code contains subtle errors in coordinate transformations, unit conversions, or array indexing. Single mistakes cascade through entire workflows in baseline LLM systems.

CLIMATEAGENT employs three complementary error recovery strategies. *Multi-candidate generation* addresses API variability: DATA-AGENT generates $m = 8$ candidate download scripts with varying interpretations (spatial bounds, temporal aggregations, variable selections) and executes them sequentially until one succeeds. *Iterative refinement* handles runtime errors: CODING-AGENT retries up to $R_{max} = 3$ times, incorporating diagnostics from previous failures, with up to 5 debug iterations per candidate. *LLM-based semantic validation* catches subtle correctness issues (wrong statistical tests, incorrect climatological periods) that produce scientifically invalid results without runtime errors.

These strategies complement each other: multi-candidate generation explores hard-to-predict parameter spaces, iterative refinement fixes implementation bugs using error feedback, and semantic validation ensures scientific correctness. Together, they enable reliable execution of complex workflows involving unpredictable APIs and intricate computations.

### 3.5 Orchestration Algorithm

Bringing together the capabilities of Coordinated Task Planning, Contextual Coordination, and Adaptive Self-Correction, Algorithm 1 synthesizes the three core capabilities into a unified orchestration process (Steps 1-5 in Figure 1). This orchestration approach ensures that complex climate science workflows are executed reliably through: (1) *context-driven coordination* enabling agents to build on prior work, (2) *multi-strategy error recovery* handling diverse failure modes, (3) *complete state persistence* supporting reproducibility and debugging, and (4) *graceful degradation* with clear failure reporting when tasks cannot be completed. These mechanisms collectively enable researchers to focus on scientific questions rather than technical implementation details.

## 4 Climate Agentic Workflow Benchmark

Another critical question for applying LLM agents for climate workflows is: *how can one systematically evaluate whether these design choices translate into reliable, high-quality scientific outputs?* This requires a benchmark that captures the authentic complexity of climate research — diverse phenomena, heterogeneous data sources, multi-step reasoning, and domain-specific correctness criteria.

Evaluating LLM-based systems for scientific workflows requires benchmarks reflecting authentic research complexity. Climate workflows demand domain-specific API knowledge, multi-step reasoning across data acquisition and analysis, integration with specialized tools, and scientifically interpretable outputs. Existing benchmarks like DataSciBench (Zhang et al., 2025a) and MASSW (Zhang et al., 2025b) focus on structured data science tasks but lack the domain complexity and real-world API integration of climate research.

To address this gap, we introduce CLIMATE-AGENT-BENCH-85, comprising 85 end-to-end workflow tasks across six climate phenomena. Each task requires executable code that retrieves data from operational APIs, processes multi-dimensional datasets, performs domain-appropriate analyses, and generates publication-quality reports. The benchmark evaluates code generation, workflow planning, error recovery under API constraints, and scientific correctness.

---

**Algorithm 1:** ClimateAgent Orchestration Workflow

---

**Input:** Climate research task $T$
**Output:** Scientific report $R$

**1** Initialize experiment directory and context $\mathcal{C}_0$;
    // Task decomposition phase
**2** $P \leftarrow \text{PLAN-AGENT.decompose}(T)$
**3** $\mathcal{C}_0.\text{plan} \leftarrow P$
    // Sequential subtask execution with error recovery
**4** **for** $i \leftarrow 1$ **to** $|P|$ **do**
      // Current subtask
**5**    $s_i \leftarrow P[i]$
      // Route to appropriate agent
**6**    $A_k \leftarrow \text{select\_agent}(s_i.\text{type})$
**7**    $\text{retry\_count} \leftarrow 0$
**8**    $\text{success} \leftarrow \text{false}$
      // Retry loop with maximum attempts
**9**    **while** $\text{retry\_count} < R_{\max}$ **and not** $success$ **do**
**10**        **if** $A_k = \text{DATA-AGENT}$ **then**
          // Strategy 1: Multi-candidate generation for downloads
**11**          $\mathcal{S} \leftarrow A_k.\text{generate\_candidates}(s_i, \mathcal{C}_{i-1}, m = 8)$
**12**          **foreach** $c \in \mathcal{S}$ **do**
**13**            $\text{result} \leftarrow \text{execute}(c, \text{exp\_dir})$
**14**            **if** $result.success$ **then**
**15**              $\mathcal{C}_i \leftarrow \text{update\_context}(\mathcal{C}_{i-1}, s_i, c, \text{result})$
**16**              $\text{success} \leftarrow \text{true}$
**17**              **break**

**18**        **else**
          // Strategy 2 & 3: Iterative refinement + semantic validation
**19**          $c \leftarrow A_k.\text{generate}(s_i, \mathcal{C}_{i-1}, \text{error\_history})$
          // Semantic validation
**20**          $\text{is\_valid} \leftarrow \text{LLM\_validate}(c, s_i, T)$
          // Runtime execution
**21**          $\text{result} \leftarrow \text{execute}(c, \text{exp\_dir})$
**22**          **if** $result.success$ **then**
**23**            $\mathcal{C}_i \leftarrow \text{update\_context}(\mathcal{C}_{i-1}, s_i, c, \text{result})$
**24**            $\text{success} \leftarrow \text{true}$
**25**          **else**
**26**            $\text{error\_history.append}(\text{result.error})$
**27**            $\text{retry\_count} \leftarrow \text{retry\_count} + 1$

    // Extract final report from coding agent output
**28** $R \leftarrow \mathcal{C}_n.\text{results}[\text{final\_report}]$
**29** **return** $R$

---

This benchmark operationalizes the three capability dimensions defined in §3 — planning, persistent context, and robustness — by translating them into measurable workflow tasks. The following sections describe how these capabilities are reflected in task design and evaluated through a unified protocol.

## 4.1 Design Principles and Capability Mapping

To empirically evaluate the three system capabilities introduced in §3, CLIMATE-AGENT-BENCH-85 is constructed around three guiding principles that mirror the design goals of CLIMATEAGENT:

- **Coordinated Task Planning.** Tasks require multi-stage decomposition — from data retrieval to analysis and visualization — so that effective workflow planning can be distinguished from ad-hoc single-step reasoning.

- **Contextual Coordination.** Many tasks contain cross-step dependencies, such as derived variables, climatological baselines, or intermediate files that must be reused downstream, testing whether an agent can maintain and propagate state across iterative reasoning and execution.

- **Adaptive Self-Correction.** Tasks involve real-world API and tool interactions (e.g., ECMWF API, TempestExtremes) under realistic parameter constraints, stressing the system's ability to detect, recover, and adapt to execution or formatting errors.

## 4.2 Task Domains and Scientific Coverage

CLIMATE-AGENT-BENCH-85 spans six climate phenomena representing diverse analysis patterns:

- **Atmospheric Rivers (AR, 15 tasks)** require computing integrated vapor transport (IVT) from multi-level wind and humidity fields, applying physical thresholds, identifying spatial regions, and tracking temporal evolution. These tests involve vertical integration, spatial pattern recognition, and trajectory analysis with wraparound coordinates.

- **Drought (DR, 15 tasks)** compute multi-timescale indices (SPI, soil moisture anomalies) requiring 30-year climatological baselines, statistical standardization handling seasonal cycles, and categorical severity visualization. These evaluate temporal aggregation, statistical edge cases (zero variance, missing data), and standard visualization conventions.

- **Extreme Precipitation (EP, 15 tasks)** analyze precipitation extremes through percentile metrics, spatial extent, and multi-day event evolution. Workflows aggregate sub-daily data, identify threshold exceedance (25–250 mm/day), and visualize progression. These assess temporal conversions, unit management, and multi-panel figures.

- **Heat Waves (HW, 10 tasks)** identify prolonged high-temperature events using multiple frameworks (absolute/percentile thresholds, duration criteria, wet-bulb temperature). These tests include multi-criteria detection, boolean logic, binary mask handling, and custom colormaps.

- **Sea Surface Temperature (SST, 15 tasks)** examine patterns, anomalies, and trends, including ENSO indices and marine heat waves. Workflows compute climatological anomalies, identify warm/cold events, and analyze spatial-temporal evolution. These require ocean-atmosphere process understanding, regional indices (Niño 3.4), and diverging color schemes.

- **Tropical Cyclones (TC, 15 tasks)** use TempestExtremes (Ullrich et al., 2021) with ERA5 and IB-TrACS data for detection, tracking, and intensity analysis. These require dynamic parameter configuration, command-line tool integration, trajectory stitching, and multi-source validation — testing tool I/O, subprocess management, and error recovery.

These domains cover atmospheric (AR, TC), hydrological (DR, EP), land surface (HW), and oceanic (SST) processes, spanning diverse scales and methodologies. AR and TC tasks test planning through multi-stage pipelines; DR and SST test context through climatological dependencies; EP and HW test robustness through complex thresholding and conversions.

## 4.3 Task Construction Methodology

**Expert-Driven Design.** Three atmospheric science graduate students from our institution (co-authors of this work) with combined expertise spanning synoptic meteorology, climate dynamics, and computational climate science systematically designed all tasks. The design process followed an iterative methodology:

- **Domain Selection**: Experts identified six phenomena that (a) represent diverse spatial and temporal scales (from daily precipitation extremes to seasonal ENSO patterns), (b) require different data sources and processing pipelines, (c) reflect common research workflows in operational and academic climate science, and (d) present varying computational and algorithmic challenges.

- **Task Diversification**: Within each domain, we designed tasks to maximize coverage of analysis patterns, data sources (ERA5, S2S forecasts, OISST, IBTrACS), spatial domains (regional to global), and temporal scales (daily to monthly). Tasks explicitly avoid redundancy — each presents unique requirements in data handling, statistical methods, or visualization conventions.

- **Real-World Grounding**: All tasks are based on actual research workflows employed in published climate studies or operational forecasting. For example, AR detection follows algorithms from Pan & Lu (2019), drought analysis implements WMO-standardized SPI methodology, and TC tracking uses established TempestExtremes configurations. This grounding ensures tasks reflect authentic scientific practice rather than artificial benchmarks.

- **Specification Refinement**: Initial task descriptions underwent multiple revision cycles to eliminate ambiguity while preserving implementation flexibility. All three experts reviewed each specification to ensure clarity, scientific accuracy, and feasibility.

This expert-driven curation ensures that each task not only reflects real-world research practice, but also targets specific capability dimensions introduced in §3. The following subsection further stratifies these tasks according to their expected planning depth, context dependence, and robustness requirements.

## 4.4 Task Complexity Stratification

We stratify tasks by workflow complexity determined by subtask steps, data sources, external tools, and algorithmic sophistication:

- **Easy Tasks (n=25, 30%)** require single-source acquisition with straightforward processing — one API call (ERA5/OISST), standard xarray operations, basic statistics, single-panel visualization. These tasks test fundamental capabilities: API parameter construction, coordinate handling, unit conversions, and basic plotting.

- **Medium Tasks (n=30, 35%)** involve multi-source data integration or multi-step workflows — coordinating multiple API calls, handling data heterogeneity (mismatched grids, resolutions, conventions), multi-stage pipelines (compute derived variable → identify events → track evolution), multi-panel figures. Medium tasks evaluate the system's ability to maintain workflow coherence across dependent steps, manage intermediate outputs, and ensure consistency.

- **Hard Tasks (n=30, 35%)** require external tool integration with dynamic parameterization — using TempestExtremes or CDO where parameters depend on runtime-discovered data characteristics, managing tool I/O, coordinating heterogeneous sources (forecasts + observations + tool outputs), implementing sophisticated algorithms (trajectory stitching, multi-criteria detection). These tasks test tool integration, subprocess management, and error recovery.

This distribution reflects realistic research workloads and aligns with capability dimensions: *easy* tasks assess planning, *medium* tasks require context maintenance, *hard* tasks challenge robustness.

## 4.5 Evaluation Protocol

To assess end-to-end system performance on CLIMATE-AGENT-BENCH-85, we design a multi-dimensional evaluation protocol aligned with the scientific communication standards of the climate community. This framework centers on a unified *report score*, which quantifies overall output quality on a 1–10 scale across four dimensions critical for scientific communication:

- **Readability**: Clarity, logical flow, accurate use of scientific terminology, and accessibility to the target audience.

- **Scientific Rigor**: Adherence to methodological standards, appropriate statistical analyses, uncertainty quantification, and validity of result interpretation.

- **Completeness**: Coverage of all task requirements, inclusion of relevant contextual information, and delivery of actionable insights.

- **Visual Quality**: Relevance and clarity of figures, appropriate visualization methods, accurate labeling, and professional presentation.

Following established practices in recent evaluation literature (Hada et al., 2024; Li et al., 2025; Zheng et al., 2023), we adopt an LLM-based judging framework for report quality assessment. This approach has demonstrated high correlation with expert human evaluation while enabling scalable and consistent scoring across large benchmarks. Our evaluator employs GPT-4o's multimodal capabilities to assess both textual content and embedded visualizations, comparing system outputs against expert-generated references. This evaluation protocol provides the foundation for the quantitative and qualitative analyses presented in §5.

## 5 Experiments

Building upon the system capabilities defined in §3 and the benchmark and evaluation framework established in §4, we now empirically examine how well CLIMATEAGENT fulfills its design objectives. Our experiments aim to answer the following three core questions:

- **Q1. Coordinated Task Planning:** Does collaborative division of workflow among specialized agents lead to higher-quality scientific reports and more complete end-to-end workflows compared with standard zero-shot LLM reasoning?

- **Q2. Contextual Coordination:** Can the system maintain cross-step dependencies and methodological consistency throughout multi-stage scientific analyses, ensuring coherent reasoning from data acquisition to interpretation?

- **Q3. Adaptive Self-Correction:** Can our error detection and recovery mechanisms enable the system to autonomously identify failures, revise its plans, and continue execution without human intervention?

To evaluate these hypotheses, we compare CLIMATEAGENT with strong baseline models on the CLIMATE-AGENT-BENCH-85 benchmark (§4), conducting quantitative, qualitative, and ablation-based analyses for each capability dimension. We first describe our experimental protocol and baseline systems (§5.1), then present quantitative results on task planning (§5.2), qualitative analysis on context coordination (§5.3), and ablation studies demonstrating adaptive self-correction (§5.4).

### 5.1 Experimental Setup

We evaluate CLIMATEAGENT on the CLIMATE-AGENT-BENCH-85 benchmark using the evaluation protocol defined in §4.5. We evaluate CLIMATEAGENT against two strong baselines:

- **GPT-5 Baseline**: A sophisticated baseline leveraging GPT-5's intrinsic reasoning capabilities with execution validation. This system employs best-of-N sampling (N=4) to generate multiple code candidates per task, executes them in a sandboxed environment, and selects the first successful solution. While capable of multi-step reasoning within single generations, this baseline lacks structured workflow decomposition, domain-specific knowledge integration, and iterative refinement mechanisms.

- **GitHub Copilot Agent Mode**: An agentic baseline utilizing GitHub Copilot's conversational capabilities for multi-turn task execution. This system leverages Copilot's advanced code generation expertise (powered by OpenAI Codex (Chen et al., 2021)) combined with its ability to maintain conversational context, execute code iteratively, and provide refinements based on execution feedback. While representing state-of-the-art general-purpose coding assistance, Copilot relies on broad programming knowledge without the domain-specific climate science expertise, structured workflow decomposition, or specialized agent coordination that characterizes our approach.

Both baseline systems utilize GPT-5 as the foundational LLM, ensuring our comparison isolates the effects of specialized agent coordination versus advanced single-model reasoning with execution validation.

Table 1: End-to-end report quality (Report Score) by domain.

| Domain | GPT-5 | Copilot | CLIMATEAGENT |
|---|---|---|---|
| Atmospheric River (AR) | 3.05 | 6.78 | **7.32** |
| Drought (DR) | 7.87 | 6.87 | **8.57** |
| Extreme Precipitation (EP) | 0.62 | 5.58 | **8.43** |
| Heatwave (HW) | 3.98 | 8.30 | **9.15** |
| Sea Surface Temperature (SST) | 4.28 | 8.10 | **8.88** |
| Tropical Cyclone (TC) | 0.00 | 2.65 | **7.85** |
| All Tasks | 3.26 | 6.27 | **8.32** |

Table 2: Overall Performance Summary on CLIMATE-AGENT-BENCH-85. All scores are averaged across all tasks on a 1-10 scale.

| System | Readability | Scientific Rigor | Completeness | Visual Quality | Report Quality |
|---|---|---|---|---|---|
| CLIMATEAGENT | **8.40** | **8.72** | **7.75** | **8.41** | **8.32** |
| Baseline (GPT-5) | 3.48 | 3.41 | 2.8 | 3.34 | 3.26 |
| Baseline (Copilot) | 6.68 | 6.89 | 5.62 | 5.87 | 6.27 |

## 5.2 Experimental Results about Task Planning

To answer Q1, we assess how coordinated task planning impacts end-to-end report generation on CLIMATE-AGENT-BENCH-85 using the setup in §4 and §5.1. For each of the six climate domains, systems must execute the full data-to-report workflow under the evaluation protocol of §4.5. We compare CLIMATEAGENT against the GPT-5 and Copilot baselines, which share the same underlying LLM but lacks explicit multi-agent decomposition and domain-specialized roles. Results are shown in Tables 1 and 2.

Under the evaluation protocol in §4.5, higher scores on Readability, Scientific Rigor, Completeness, Visual Quality, and overall Report Quality can all be interpreted as downstream consequences of more effective task planning: more coherent decomposition and delegation should yield clearer narratives (Readability), more appropriate methodological choices (Scientific Rigor), fuller coverage of required steps (Completeness), and better targeted figures (Visual Quality).

**Experiment Results and Discussions.** We summarize the main results and discussion below:

- **Overall Impact of Coordinated Planning.** Across all 85 tasks, CLIMATEAGENT achieves the highest average Report Quality (**8.32** vs. 6.27 for Copilot and 3.26 for GPT-5; Table 2), indicating that explicit division of labor among specialized agents leads to substantially better end-to-end reports than single-model reasoning with execution validation. These gains are consistent across all six domains (Table 1), with particularly large margins in complex, multi-stage settings such as Extreme Precipitation (8.43 vs. 5.58 vs. 0.62) and Tropical Cyclones (7.85 vs. 2.65 vs. 0.00), where baselines frequently fail to complete multi-step analyses.

- **Completeness and Scientific Rigor as Planning Outcomes.** Beyond overall Report Quality, CLIMATEAGENT also substantially outperforms baselines on **Completeness** (**7.75** vs. 5.62 vs. 2.80) and **Scientific Rigor** (**8.72** vs. 6.89 vs. 3.41). Interpreted through the lens of Q1, higher Completeness reflects that the PLAN-AGENT decomposes each task into concrete subgoals and assigns them to specialized DATA-AGENTs and CODING-AGENTs, reducing missing figures, truncated analyses, and omitted discussion that commonly occur in the GPT-5 and Copilot baselines. Higher Scientific Rigor indicates that coordinated planning encourages a more disciplined methodological choices — selecting appropriate datasets, applying physically meaningful aggregations, and justifying parameter choices in a way

that aligns with expert expectations — rather than the ad-hoc, inconsistent methods often observed in single-model workflows.

- **Readability and Visual Quality as Planning Effects.** Coordinated planning also improves how results are communicated. As shown in Table 2, CLIMATEAGENT attains higher **Readability** (8.40 vs. 6.68 vs. 3.48) and **Visual Quality** (8.41 vs. 5.87 vs. 3.34) than both baselines. From the perspective of Q1, these gains arise because the PLAN-AGENT explicitly anticipates the narrative and visual requirements of each task — specifying which diagnostics, figures, and explanatory sections are needed — and delegates them to specialized agents. This leads to reports with clearer structure, better-matched figures, and consistent labeling that directly support the planned analytical storyline. In contrast, single-model baselines often generate plots and text in a more opportunistic, step-by-step manner, so narrative flow, figure selection, and captioning drift away from the original task specification, lowering both readability and visual quality, despite the successful execution of some individual steps.

**Summarized Answer to Q1.** Taken together, the improvements in Report Quality, Completeness, Scientific Rigor, Readability, and Visual Quality demonstrate that coordinated task planning is a key determinant of end-to-end performance on CLIMATE-AGENT-BENCH-85. By explicitly decomposing workflows and distributing responsibilities across specialized agents, CLIMATEAGENT produces higher-quality scientific reports and more complete end-to-end workflows than strong GPT-5–based single-model baselines. These quantitative findings support Q1, confirming that collaborative division of labor among specialized agents yields systematic gains over advanced single-model reasoning with execution validation.

### 5.3 Experimental Results about Context Coordination

To answer Q2, we complement the quantitative report scores with a qualitative analysis of end-to-end workflows. Because each report is the outcome of a multi-step climate analysis pipeline, any miscoordination between stages (e.g., inconsistent data sources, mismatched parameter choices, or failed intermediate steps) directly degrades report quality or even prevents report generation altogether. While the scores in §5.2 show how well each system performs overall, they do not reveal why a system succeeds or fails. By qualitatively examining generated reports and their underlying execution traces, we can assess whether each system maintains the necessary cross-step dependencies and methodological consistency that define contextual coordination. Figure 2 presents representative outputs across the six climate domains, which we use to analyze how each system maintains (or fails to maintain) contextual coordination in practice.

**Experiment Results and Discussions.** We enumerate the key observations and discussion below:

- **Baseline Competence in Simpler Domains.** For simpler domains such as Drought (DR) and Heatwave (HW), both baselines generally produce reasonable figures and narratives: the underlying workflows involve fewer stages, simpler data choices, and more direct mappings from task description to code. In these settings, Copilot substantially outperforms GPT-5, reflecting the benefit of iterative code refinement and execution feedback for correcting obvious bugs and filling in missing steps. However, even in these easy domains, baseline reports sometimes exhibit mild inconsistencies in variable naming, axis labeling, or temporal aggregation, foreshadowing the more severe coordination failures that emerge as workflow complexity increases. In contrast, CLIMATEAGENT maintains a stable mapping between task specification, data selection, and visual presentation, even in these simpler cases, indicating that contextual coordination mechanisms are active across the full task spectrum.

- **Monolithic Scripts.** As workflows become more complex in Atmospheric River (AR) and Sea Surface Temperature (SST) tasks, baseline systems increasingly default to monolithic scripts with limited explicit context management. Qualitative inspection of execution traces shows a recurring pattern: changes made late in the script to fix an exception or adjust a plot overwrite earlier logic without updating dependent steps, so the final code no longer matches the original analytical intent. This leads to misaligned preprocessing, inconsistent temporal or spatial domains across figures, and missing climatological references in SST tasks. CLIMATEAGENT, by contrast, decomposes these workflows into explicit substeps with clearly defined inputs and outputs, ensuring that updates to one stage (e.g., data filtering or anomaly com-

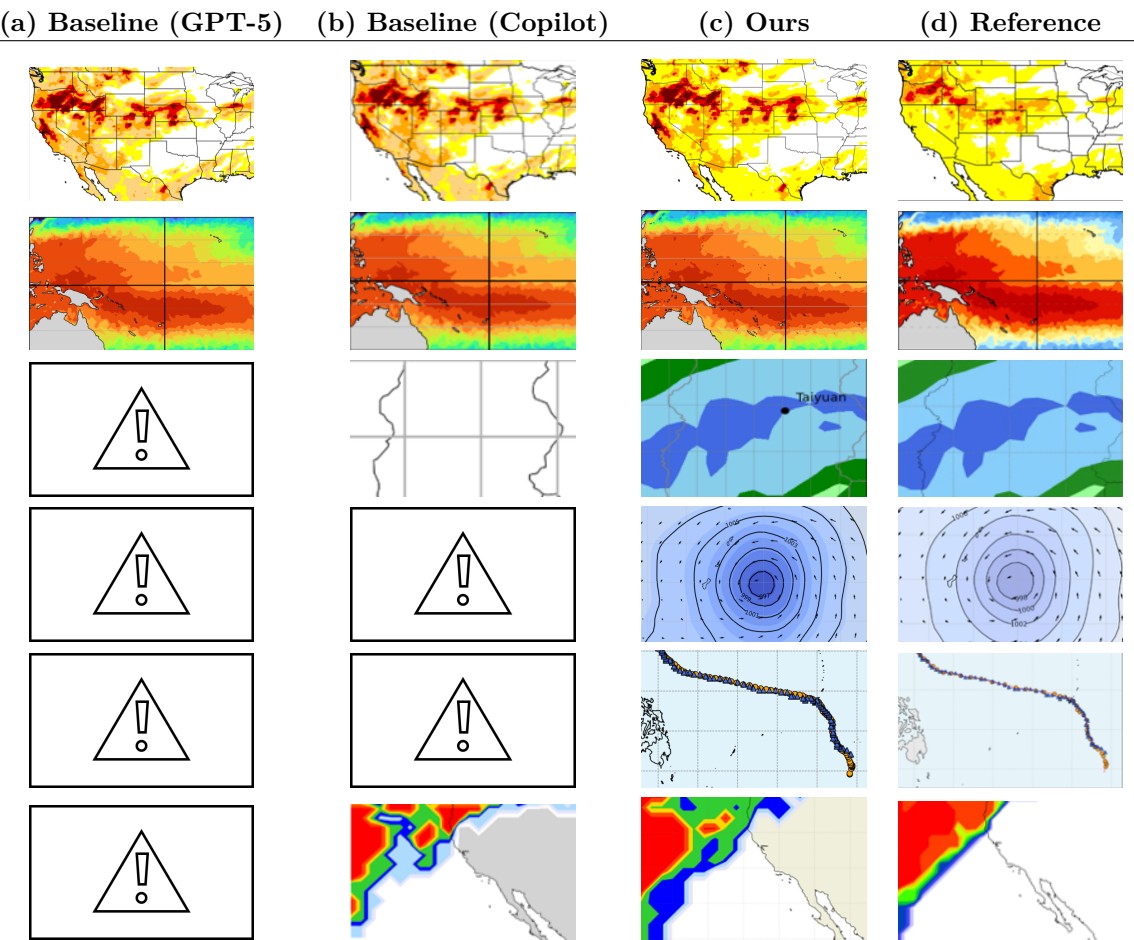

Figure 2: Qualitative comparison of generated figures for representative tasks. Each row corresponds to a climate task: (1) Drought (DR), (2) Sea Surface Temperature (SST), (3) Extreme Precipitation (EP), (4-5) Tropical Cyclone (TC), (6) Atmospheric River (AR). Columns: (a) Baseline (GPT-5), (b) Baseline (Copilot), (c) Ours, (d) Golden answer.

putation) can be propagated to downstream analyses and visualizations, thereby preserving cross-step dependencies required for contextual coordination.

- **Untracked External Processes.** In demanding domains such as Tropical Cyclone (TC) and Extreme Precipitation (EP), baseline failures are often driven by how external tools and processes are orchestrated. When GPT-5 or Copilot spawns new processes (e.g., calling TempestExtremes for TC tracking), return codes and intermediate outputs are rarely tracked rigorously. Downstream code frequently proceeds as if upstream stages had succeeded, attempting to read missing files, operate on partially written datasets, or plot fields that were never computed. This results in runtime crashes, empty or nonsensical visualizations, and incomplete reports, especially in TC tasks that rely on multi-step external toolchains. In contrast, CLIMATEAGENT treats each external invocation as an explicit workflow step, recording return statuses, verifying that expected artifacts exist and are well-formed, and halting or repairing the pipeline when upstream tools fail, thereby maintaining coherent execution state across process boundaries.

- **Missing Intermediate Validation.** Across all domains, these issues are compounded by a pervasive lack of systematic validation of intermediate results in the baselines. Figures are often generated without checking whether the underlying data satisfy task requirements (e.g., correct spatial subset, sufficient temporal coverage, or non-degenerate statistics), leading to qualitatively poor or even empty plots when earlier steps quietly fail or return trivial outputs. A misconfigured data request or misaligned coordinate system at the beginning of the workflow can thus cascade into misleading or uninformative visualizations

at the end, with no mechanism to detect or correct the deviation. CLIMATEAGENT mitigates this cross-domain failure mode by embedding validation hooks throughout the workflow — checking the dataset metadata, asserting non-empty and physically plausible fields, and aligning coordinate systems before plotting — so that each stage both consumes and produces well-validated context, reinforcing contextual coordination from data acquisition through to final report generation.

**Summarized Answer to Q2.** These findings confirm that **Contextual Coordination** (Q2) is the critical differentiator for robust scientific workflows. The analysis reveals that baseline failures in complex domains are rarely due to local coding errors, but rather a structural inability to maintain state across long horizons and process boundaries. CLIMATEAGENT overcomes this by enforcing explicit, validated handoffs between agents, effectively treating intermediate artifacts as contracts. This persistent state management ensures that downstream execution remains strictly conditioned on upstream results, preventing the context drift that causes single-model baselines to lose the analytical thread in multi-stage tasks.

### 5.4 Ablation and Case Studies for Adaptive Self-Correction

We now turn to answer Q3 — the system's ability to autonomously identify failures, revise its plans, and continue execution without human intervention. This section analyzes common baseline failure modes and demonstrates how CLIMATEAGENT achieves robustness through multi-layered detection, recovery, and adaptive replanning mechanisms. To validate this capability, aggregate performance metrics are insufficient; we must instead isolate the specific mechanisms of failure and recovery. Therefore, we adopt a two-pronged approach: (1) an ablation analysis of the distribution of errors in baselines to define the failure modes our system must overcome, and (2) a case study tracing a complex recovery loop to demonstrate the self-correction mechanism in action. Beyond these representative self-correction case studies, we provide a dedicated failure-case analysis (with logs, artifacts, root causes, and mitigations) in Appendix E and component-level ablation analysis in Appendix D.

**Ablation Analysis.** We systematically classified GPT-5 baseline errors across 35 failed tasks, identifying six primary categories (Table 3): Data/Array Shape or Key Errors (26%), Data Request Errors (17%), Syntax/Indentation Errors (11%), Timeout Errors (11%), Type Errors (11%), and Miscellaneous (23%). These failures predominantly result in incomplete or absent report generation, demonstrating direct LLM code synthesis limitations without system-level safeguards.

Table 3: Summary of Error Categories and Their Counts

| Error Category | Count |
|---|---|
| Data/Array Shape or Key Error | 9 |
| Data Request Error | 6 |
| Syntax/Indentation Error | 4 |
| Timeout Error | 4 |
| Type Error | 4 |
| Miscellaneous | 8 |

Our system incorporates several architectural and prompt-based interventions that directly address the failure modes observed in the baseline. Figures 3–6 illustrate how specialized DATA-AGENT and CODING-AGENT components validate dataset metadata, enforce typing, and iteratively repair code before execution, eliminating the majority of such errors. We summarize the key findings below:

- **Data/Array Shape or Key Error:** As illustrated in Figure 3, baseline approaches often fail due to incorrect assumptions about data structure or mismatched array dimensions. Our system addresses these issues: DATA-AGENTS extract and validate dataset metadata before code generation, ensuring that only available variables and correctly shaped dimensions are used. Furthermore, CODING-AGENTS perform LLM-based code validation to verify data access patterns before execution, systematically preventing such shape and key errors.

**Baseline GPT-5 Error Example**

```
# Incorrect boolean mask shape
ivt_selected = ivt[:, mask]
```

**Our System Success**

```
# Ensures mask shape matches data shape
mask = (ivt > threshold)
# 1D selection
ivt_selected = ivt[mask]
```

Figure 3: Comparison of array indexing: the baseline code fails due to a shape mismatch in boolean indexing, while our system validates shapes and applies correct masking.

**Baseline GPT-5 Error Example**

```
# Incorrect date range format
# in API request
request = {
    ...
    "date": f"{req_start}/to/{req_end}",
    ...
}
```

**Our System Success**

```
# Validates date range for each month
request = {
    "year": str(year),
    "month": month,
    "day": days,
}
```

Figure 4: Comparison of ERA5 data request formatting: the baseline code fails due to an invalid date range string, while our system programmatically generates and validates correct request parameters, preventing API errors.

**Baseline GPT-5 Error Example**

```
# Syntax error: unmatched '}'
dtxt += f} {bearing}"
```

**Our System Success**

```
# No syntax or indentation errors.
```

Figure 5: Comparison of syntax handling: the baseline code fails with a syntax error due to an unmatched brace, while our system's coding agent ensures only syntactically valid code is executed.

**Baseline GPT-5 Error Example**

```
# TypeError: not all arguments
# converted during string formatting
df["LON"].apply(wrap_lon_to_360)
```

**Our System Success**

```
# Convert LON to numeric
# before modulo operation
pd.to_numeric(df_filt["LON"])
df_filt["LON"] % 360
```

Figure 6: Comparison of longitude alignment: the baseline code fails when non-numeric values are present, while our system ensures type safety before applying arithmetic operations.

- **Data Request Error:** As illustrated in Figure 4, baseline approaches fail when interacting with ERA5 or ECMWF APIs, often due to improperly formatted requests or insufficient parameter validation. Our DATA-AGENTs employ automated metadata extraction and validation routines before code generation, utilizing LLM-driven logic to dynamically select valid variables, date ranges, and request options, referencing up-to-date dataset metadata to construct compliant API calls.

- **Syntax/Indentation Error:** Figure 5 highlights how baseline LLM-generated code frequently encounters syntax and indentation problems that halt execution. Our CODING-AGENTs proactively check for such errors before running any code, using diagnostic feedback to iteratively refine and correct the code, ensuring that only error-free scripts proceed to execution.

- **Type Error:** As demonstrated in Figure 6, baseline code often fails due to improper handling of data types. Our system integrates type validation and conversion directly into the workflow, with the CODING-AGENT automatically checking and enforcing correct data types — guided by both prompt instructions and LLM-based code review — before any computation is performed.

**Case Study.** The tropical-cyclone task focusing on Typhoon Noru (SID: 2022264N17132) demonstrates the full value of our robustness mechanisms. This task compares the observed historical track from the IBTrACS dataset against a simulated track generated using ERA5 reanalysis data, producing a meteorological map visualizing both tracks alongside a summary report quantifying track differences and forecast accuracy.

The baseline GPT-5 code fails to complete the workflow, terminating with:

*ERROR: Failed to select longest track. single positional indexer is out-of-bounds*

This failure is due to incomplete or improperly parsed track data.

In contrast, our system executes a robust, agent-based workflow:

- PLAN-AGENT: Decomposes the TC analysis into 10 explicit subtasks, including: (1) reading and processing IBTrACS data, (2) determining ERA5 download parameters, (3) downloading ERA5 reanalysis data, (4) computing TempestExtremes detection parameters, (5) running DetectNodes, (6) running StitchNodes, (7) extracting the longest simulated track, (8) visualizing meteorological fields, (9) extracting central pressure, and (10) generating the final Markdown report.

- DATA-AGENT: Utilizes metadata extraction and LLM-guided parameter selection to construct valid ERA5 API requests. The agent automatically identifies the correct dataset (`reanalysis-era5-single-levels`) and validates all required parameters (such as date ranges and area) before making the API call.

- CODING-AGENT: Implements robust scripts for each downstream subtask, including dynamic parameter computation, file validation, and error handling. Each script checks for the existence and integrity of its outputs before passing control to the next step, ensuring that failures are caught early and reported with actionable diagnostics. Finally, the agent compiles the Markdown report, embedding the meteorological field plot and central pressure value.

**Summarized Answer to Q3.** These results substantiate **Adaptive Self-Correction (Q3)**. The proper handling of errors, combined with the qualitative success in the Typhoon Noru case study, demonstrates that robustness in scientific agents cannot be achieved by LLM reasoning alone. Instead, it requires a system architecture that treats code generation as a hypothesis to be validated — using metadata constraints, static analysis, and execution feedback to autonomously correct the "hallucinations" that otherwise break long-horizon scientific workflows.

## 6 Conclusion

We have introduced CLIMATEAGENT, an autonomous multi-agent system designed to orchestrate complex climate science workflows from high-level user prompts to comprehensive scientific reports. By leveraging a hierarchical architecture of specialized LLM-based agents, i.e., each responsible for planning, data acquisition, analysis, and visualization, our system addresses the limitations of generic code-generation models and static scripting approaches. Extensive evaluation on the CLIMATE-AGENT-BENCH-85 benchmark demonstrates that CLIMATEAGENT substantially outperforms advanced single-model baselines across all domains, particularly in tasks requiring multi-step reasoning, robust error handling, and domain-specific knowledge. Our results highlight the effectiveness of modular agent specialization, dynamic error recovery, and context-aware orchestration in enabling reliable, end-to-end automation of climate research workflows. This work advances the state-of-the-art in scientific workflow automation and paves the way for more accessible, efficient, and reproducible climate science.

**Acknowledgments**

This research is supported by the Hong Kong Research Grants Council's collaborative research funds (project no. C6032-21G), general research fund (project nos. 16300424 & 16215820). M. Lu also acknowledges the support by the Otto Poon Centre for Climate Resilience and Sustainability at HKUST. This work is part of the United Nations Educational Scientific and Cultural Organization (UNESCO) International Decade of Sciences for Sustainable Development (2024–33) and contributes to the Seamless Prediction and Services for Sustainable Natural and Built Environment (SEPRESS) Program (2025–32), an initiative endorsed under this global framework.

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

# A    Agent Details

## A.1    Plan-Agent: Task Decomposition and Agent Assignment

The PLAN-AGENT is responsible for decomposing high-level user tasks into a sequence of actionable subtasks, each assigned to a specialized agent within the CLIMATEAGENT system. Leveraging large language models, the PLAN-AGENT generates detailed, chronologically ordered plans that preserve all scientific and technical requirements from the original user prompt. Each subtask includes explicit instructions, file paths, parameter values, and workflow conventions to ensure downstream agents can execute their roles unambiguously and reproducibly.

**Prompt Engineering.** The PLAN-AGENT constructs prompts that enumerate available agents (data download, programming, visualization), file system conventions, and dataset constraints. It instructs the LLM to retain all relevant details from the user task, specify agent assignments, and output the plan as a structured JSON list. This approach ensures that each subtask is both specific and actionable, minimizing ambiguity and error propagation.

---

**Plan-Agent LLM Prompt**

You are a planning agent for a modular climate forecasting/reporting system. The system has four main agents:

- `cdsapi_download_agent`: downloads climate data only using the cdsapi library (for Copernicus Data Store datasets).

- `data_download_agent`: downloads climate data only using ecmwf-api-client library (for ECMWF S2S dataset only).

- `programming_agent`: processes, analyzes, and computes on climate data (but does not download data). It should generate plots/graphs that will be used in the final report.

- `visualization_agent`: generates the final report in Markdown format, including all requested plots, visualizations, and human-friendly interpretation. The final report generation should always be handled by the visualization_agent.

**File System & Data Conventions**

- Only download agents may write files to `data/`.

- CODING-AGENTs must write all outputs (processed data, intermediate results, analysis outputs, figures, etc.) to `code_output/`.

- Visualization agents should read from `code_output/`.

- All user-provided data must be referenced under `../user_provided_data/` (relative to the task root).

**Critical:** When generating subtasks, you MUST preserve as much detail as possible from the original user task prompt. For each subtask, explicitly include all relevant parameters, file paths, scientific logic, and requirements from the original prompt. Do not summarize or omit details. If the original prompt specifies variables, thresholds, file formats, coordinate conventions, or workflow steps, these MUST be included in the subtask description. The goal is for each subtask to be as actionable and unambiguous as possible for downstream agents.

**Planning Guidelines**

- Break down the main task into small, logical, and chronologically ordered steps.

- For each subtask, specify the agent, the action, and all required details (parameters, files, scientific logic, etc.).

---

- Be specific and unambiguous: name required data sources, variables, parameters, and outputs precisely.

- If a subtask involves generating a plot/graph, specify the type of plot and the data to be used.

- If a subtask is for the visualization_agent, make sure the description clearly states that it is for the final report and should include all required plots and interpretation.

- Only include agent types relevant to the task (do not include unused agent types).

**Example of Generated Plan.** Below is an example of a plan generated by the PLAN-AGENT for an atmospheric river (AR) detection workflow. Each subtask is assigned to the appropriate agent and includes a detailed description of the required actions, parameters, and output conventions.

---

**Plan-Agent Output Example**

**Subtasks:**

1. **Create configuration file and directory structure for AR detection workflow**
   *Agent: programming_agent*
   Create `code_output/config.py` with constants and paths, and ensure all directories exist. Write `run_metadata.json` in `code_output/outputs_ar_freq` with the configuration used.

2. **Download ERA5 pressure-level q, u, v for 2022-12-19 to 2022-12-25 at 00:00 UTC**
   *Agent: cdsapi_download_agent*
   Use cdsapi to request 'reanalysis-era5-pressure-levels' data for specific humidity, u and v wind components, at specified pressure levels and times. Save output to `data/era5_raw/era5_pl_uvq_20221219-20221225_0000.grib`.

3. **Compute daily IVT on a 1.5° x 1.5° grid for each date in DATE_RANGE**
   *Agent: programming_agent*
   Script: `code_output/compute_ivt.py`. Load downloaded data, regrid to 1.5° grid, compute IVT, and save daily NetCDF files with compression.

4. **Load monthly IVT threshold and run AR detection per day (extract, rotate, paths, trajectories, metrics)**
   *Agent: programming_agent*
   Script: `code_output/detect_ar_pathways.py`. Load threshold, apply rotation, compute anomaly, extract paths, generate trajectories, and save outputs in hierarchical date folders.

5. **Weekly aggregation of daily pathway presence and create frequency map image**
   *Agent: programming_agent*
   Script: `code_output/visualize_frequency.py`. Aggregate daily pathway files, compute weekly frequency, subset to US West Coast, save NetCDF, and plot frequency map using Cartopy.

6. **Generate final Markdown report with the exact reference text and embedded image**
   *Agent: visualization_agent*
   Create `code_output/report.md` referencing the frequency map image and containing the required summary and bullet points.

## A.2 Data-Agents: Prompt Engineering and Metadata Extraction

The DATA-AGENTs in CLIMATEAGENT, namely ECMWF-AGENT and CDSAPI-AGENT, are responsible for generating robust Python scripts to download climate datasets from ECMWF S2S and Copernicus Data Store (CDS), respectively. These agents leverage large language models (LLMs) for code synthesis and employ automated metadata extraction using browser automation to ensure parameter validity and dataset availability.

**Prompt Engineering.** Both agents utilize carefully designed prompts to instruct the LLM in generating executable download scripts. The prompts include explicit requirements for code structure, error handling, output directory usage, and metadata integration.

**ECMWF-Agent Prompt Example.** We attach the agent prompt for ECMWF below:

---

**ECMWF-Agent LLM Prompt**

You are an expert in ECMWF S2S data download. Given the following task, parameter info, and metadata JSONs, write a Python script that downloads the required data using the ecmwf-api-client library.
- Use only the available options in the metadata JSONs (see file names and their content summaries below).
- The purpose of these metadata JSONs is to reduce errors in the API calls of the generated Python code.

General Requirements for the Code:

- The code must be modular, well-structured, and include clear, descriptive comments explaining each step and function.

- Follow Python best practices for readability, maintainability, and efficiency.

- Use appropriate scientific/data libraries (e.g., numpy, pandas, matplotlib, xarray, etc.).

- All necessary imports must be included at the top of the script.

- All code should be directly executable, with all necessary fields and values filled in.

- The script must be runnable as: python [code_name].py (with no arguments). Do not require or parse any command-line arguments.

- All downloaded data files must be saved in the 'data' subdirectory of the current task folder. Do not save data files elsewhere.

- Do NOT use 'experiments/user_provided_data' for any downloaded data outputs. That directory is reserved for user-provided data only.

- All relative paths should be constructed relative to the directory the code is running. Don't use absolute paths.

IMPORTANT: For efficiency, always batch all required parameters and levels into a single API call using '/'-separated lists for 'param' and 'levelist'. Do NOT make a separate retrieve call for each parameter or level unless required by the API. Only loop over forecast type (cf/pf) or origin if absolutely necessary.

Available parameters (with codes): `{AVAILABLE_PARAMS_TEXT}`

---

IMPORTANT: Always use the correct 'origin' code for the requested model/database.

Special Instructions for Data Downloading:

- If the subtask involves data download, you must use the `ecmwf-api-client` library or the provided download tool.

- Only include the required fields below in the API call (do not add any others, especially 'area'):
    - class: s2
    - dataset: s2s
    - date: <date range for real-time|model version date for hindcast>
    - expver: prod
    - levelist: <level range> (only for pl, omit for sfc)
    - levtype: <sfc|pl> (if requesting both, call API separately for each)
    - model: glob
    - origin: <origin> (e.g. anso, ecmf, kwbc)
    - param: <parameter> (if requesting multiple parameters, call API separately for each)
    - step: <step range> (use a '/'-separated list of step values)
    - stream: <enfo|enfh> (enfo: realtime, enfh: hindcast)
    - time: '00:00:00'
    - type: <cf|pf> (cf: control, pf: perturbed)
    - target: <target file name>
    - hdate: <yyyy-mm-dd> (only for hindcast, specify a list of hindcast initialization dates)
    - number: <number of ensemble members> (only for perturbed hindcast, use '/'-separated list for multiple members)

- Do not include any other fields.

Guideline for Setting 'date', 'hdate', and 'step' Fields in Requests:

1. Real-Time Forecast Setting: Use the operational model version available on the date.

2. Hindcast (Reforecast) Setting: Use the most recent available model version date strictly before the requested date.

3. 'step' field: For daily-averaged parameters, use hour ranges representing 24-hour periods; for instantaneous/accumulated parameters, use single time steps.

After Downloading the Data:

- Create/Update a README.md file in the data directory, listing all downloaded files and their descriptions.

- For GRIB files, use `cfgrib` to extract and include metadata summaries in the README.

- For other file types, provide appropriate previews or summaries.

Metadata JSONs (file name, description, and content preview): `{meta_block}`

**Task description: {task_description}**

Return only the Python code, with all explanations and context provided through code comments. Do not include any narrative or markdown outside the code block.

**CDSAPI-Agent Prompt Example:** We attach the agent prompt for CDSAPI below:

---

**CDSAPI-Agent LLM Prompt**

You are an expert in CDS (Copernicus Data Store) data download. Given the following task, write a Python script that downloads the required data using the cdsapi library.

- The code must be modular, well-structured, and include clear, descriptive comments explaining each step and function.

- Use only the required fields for the cdsapi call (see https://cds.climate.copernicus.eu/api-how-to for reference).

- All necessary imports must be included at the top of the script.

- All code should be directly executable.

- The script must be runnable as: python [code_name].py (with no arguments). Do not require or parse any command-line arguments.

- All downloaded data files and the README.md must be saved in the directory: [DATA_DIR], which will be the current working directory when the script is run.

- The script must use the current working directory (`os.getcwd()`) or a provided variable for all output paths.

- After downloading, create or update a README.md file in the data directory, listing the files and a brief description of their contents.

- If the dataset or variable is not available, the script should print a clear error message.

- Note that the `cdsapi.Client` only supports the `retrieve` method.

- At the end of the script, print to stdout a single line containing a JSON array of the absolute paths of all files that were downloaded by the script. For example: `print(json.dumps(["/path/to/file1", "/path/to/file2"]))`

- Do not print anything else to stdout after this line.

**Task description:** `{task_description}`

**Metadata:** `{metadata_str}`

Return only the Python code, with all explanations and context provided through code comments. Do not include any narrative or markdown outside the code block.

---

**Metadata Extraction via Chrome/Selenium.** To dynamically identify available datasets, variables, and valid parameter ranges, both agents use Selenium with Chrome in headless mode. The agent navigates to the relevant data portal, interacts with web forms, and parses metadata (e.g., from JavaScript objects or HTML elements). Extracted metadata is saved as JSON and provided to the LLM as context for code generation, ensuring that only valid options are used in download requests.

**Outputs.** The agents produce several outputs for each download task:

- **Generated Python Script:** A modular, well-commented script that downloads the requested data using validated parameters.

- **README.md:** A summary file listing downloaded files and their descriptions.

- **Metadata JSON:** A record of available dataset options and parameters.

---

**Example: README.md file**

ERA5 Pressure-Level Data Raw Download
**Dataset:** reanalysis-era5-pressure-levels
**Variables:** specific_humidity, u_component_of_wind, v_component_of_wind
**Pressure levels (hPa):** 1000, 925, 850, 700, 500, 300, 200
**Time:** 00:00 UTC
**Date range:** 2022-03-24 through 2022-03-30
**Format:** GRIB
**File path:** `data/era5_raw/era5_pl_uvq_20220324-20220330_0000.grib`

These data are native ERA5 pressure-level fields suitable for IVT computation.

---

**Discussion.** By combining LLM-driven code generation with automated metadata extraction, the DATA-AGENTs reduce errors due to invalid parameters and improve reproducibility. This approach enables the system to adapt to evolving data portals and ensures that download scripts remain robust and up-to-date.

### A.3 Coding-Agent (Programming): Data Processing and Analysis

The CODING-AGENT (Programming) is responsible for generating Python code to perform analysis and processing subtasks within the ClimateAgent workflow. This agent leverages large language models (LLMs) to synthesize modular, well-documented scripts for scientific data analysis and post-processing, based on the main user task and specific subtask descriptions.

**Prompt Engineering.**   The agent constructs detailed prompts for the LLM, specifying requirements such as code modularity, use of scientific libraries (e.g., numpy, pandas, xarray, matplotlib), and strict file system conventions. The prompt instructs the LLM to avoid data download operations (handled by dedicated agents), save all outputs in the `code_output/` directory, and update or create a `README.md` file describing generated outputs. Debugging instructions and error messages are included in the prompt when code regeneration is required.

---

**Coding-Agent (Programming) LLM Prompt**

You are an expert Python programmer and agent developer. You are programming for predicting and forecasting atmospheric phenomena in climatology. Downloads from ECMWF datasets are already handled by the DATA-AGENTs. Do not write code to download climate data.
General Requirements for the Code:

- The code must be modular, well-structured, and include clear, descriptive comments explaining each step and function.

- Follow Python best practices for readability, maintainability, and efficiency.

- Use appropriate scientific/data libraries (e.g., numpy, pandas, matplotlib, xarray, etc.).

- All necessary imports must be included at the top of the script.

- All code should be directly executable.

- All output files must be saved in the `code_output/` directory under the current task root.

- Do not write any files to the `data/` directory.

- All user-provided data must be loaded from the directory `../user_provided_data/`.

- For every output file generated, also create or update a `README.md` file in the output directory describing the outputs.

- Plotting tip: Always place your colorbar or legend outside the main plot area and use `tight_layout` or `constrained_layout` for spacing.

Write a Python script that executes the given **subtask**. At the end of the script, include a test script to validate the generated output. Return only the Python code, with all explanations and context provided through code comments. Do not include any narrative or markdown outside the code block.
**Main task:** {main_task}
**Subtask:** {subtask}
**Previous subtasks and codes:** {previous_codes}
**Directory structure:** {dir_tree}
**README.md summary:** {readme_summary}

---

**Workflow and Error Recovery.**   For each subtask, the agent generates multiple candidate scripts, validates their syntax, and ranks them using LLM-based code review for correctness, robustness, and clarity. If all candidates fail, the agent enters a debug loop, providing error messages and previous code to the

LLM for iterative refinement. Successful code execution triggers automatic updates to the `README.md` file, documenting outputs and code changes.

**Output.** The following is an example of a `README.md` file automatically generated by the CODING-AGENT (Programming) after completing a subtask for weekly sea surface temperature (SST) analysis. This file documents the produced figures and reports, describes the contents and projection details, and provides metadata for reproducibility and further analysis.

---

**Example: README.md for Analysis Output**

**Weekly SST and SST Anomaly Map**
This directory contains the high-resolution two-panel map showing:

- Weekly mean Sea Surface Temperature (SST) for June 12–18, 1997

- Weekly mean SST Anomalies for the same period

**File:**

- `weekly_sst_anomaly_map.png`: Two-panel PNG figure [12×8 in, 300 DPI]

The map uses a PlateCarree projection (central_longitude $= -155°$, extent 120°E–290°E, $\pm 30°$ latitude), with bold black lines at the equator and 180° meridian, and coastlines/land shaded in gray.

---

**Discussion.** By automating code generation, validation, and debugging, the programming agent streamlines scientific analysis and ensures reproducibility. Its design enforces strict conventions for output management and documentation, facilitating transparent and collaborative climate research workflows.

### A.4 Coding-Agent (Visualization): Report Generation and Plotting

The CODING-AGENT (Visualization) automates the creation of scientific reports and visualizations as the final step in the ClimateAgent workflow. This agent synthesizes Markdown documents, figures, and summary files by leveraging large language models to interpret analysis outputs and generate publication-quality content. It ensures that all results are saved in standardized directories and that each output is accompanied by a descriptive `README.md` file for reproducibility.

**Prompt Engineering.** The visualization agent constructs prompts that specify the required report structure, output file conventions, and documentation standards. The prompt instructs the LLM to:

- Generate all requested plots and Markdown content as specified in the subtask.

- Save all outputs (figures, CSVs, Markdown) in the `code_output/` directory.

- For every output file, create or update a `README.md` in its directory describing the outputs.

- Ensure the final report is a Markdown file named `final_report.md` in `code_output/`, containing all relevant analysis and images.

- Load any user-provided data from the standardized directory `experiments/user_provided_data/`.

- Return only the Python code, with all explanations and context provided through code comments, and no extraneous markdown or narrative.

---

**Coding-Agent (Visualization) LLM Prompt**

You are responsible for fully executing the following visualization/reporting subtask. Generate all required plots, Markdown, and outputs as specified, and ensure all results are saved and documented as described below.
Requirements:

- Assume code is run from the task root (`[output_dir]`).

- Save all outputs (figures, CSVs, Markdown) in `code_output/`.

- For every output file, create or update a `README.md` in its directory describing the outputs.

- The final output must be a Markdown report named `final_report.md` in `code_output/`, containing all relevant analysis and images.

- All user-provided data must be loaded from `experiments/user_provided_data/`.

- Return only the Python code, with all explanations and context provided through code comments. Do NOT include any narrative, markdown, or code block markers outside the code.

**Subtask to execute:** {subtask}
**Main Task:** {main_task}
**Context:** Directory structure, previous subtasks, previous codes, and key README.md summaries.

---

**Workflow and Error Recovery.** For each visualization subtask, the agent generates multiple candidate scripts, validates their execution, and iteratively refines code in response to errors. It gathers context from previous analysis outputs, directory structure, and documentation to ensure consistency and completeness in the final report.

**Output Examples.** Below are two examples of `final_report.md` files automatically generated by the CODING-AGENT (Visualization) after completing different visualization subtasks. These reports demonstrate the agent's versatility in synthesizing diverse scientific findings — from multi-day precipitation events to

tropical cyclone meteorological analyses — into publication-ready markdown documents with embedded figures, spatial analysis, and narrative summaries for transparent communication and reproducibility.

**Example: final_report.md for Extreme Precipitation Event**

**Extreme Precipitation Event in the Greater Bay Area (2023-09-05 to 2023-09-10)**
**Introduction**
This report summarizes an exceptional precipitation event that affected the China Greater Bay Area between September 5 and 10, 2023. Hourly total precipitation (`tp`) data were obtained from the ERA5 reanalysis via the Copernicus Climate Data Store and aggregated to daily totals (mm). Spatial analysis and visualization were performed on the native ERA5 grid, highlighting regions exceeding common thresholds (25, 50, 100, 250 mm) and tracking the evolution of rainfall cores.
**Multi-Panel Precipitation Map**

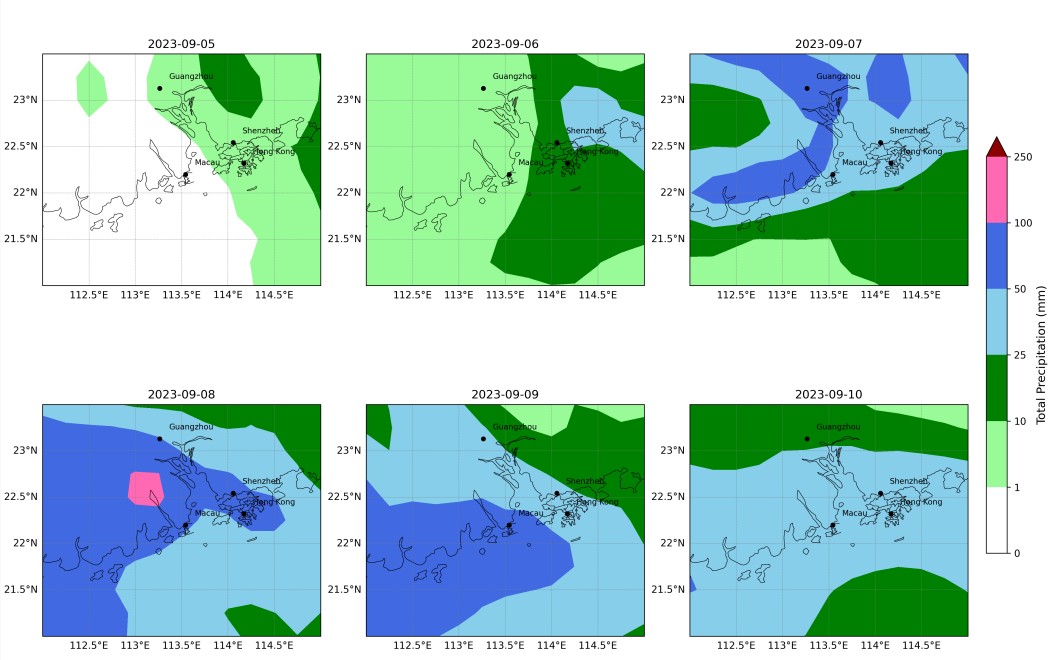

*Figure 1.* Daily total precipitation in the Greater Bay Area from September 5 to 10, 2023. Stepped color intervals (0, 1, 10, 25, 50, 100, 250 mm) illustrate rainfall intensity. Major cities (Guangzhou, Shenzhen, Hong Kong) are marked for reference.
**Event Evolution Narrative**
**Onset (Sep 5–6):** The event began on September 5 with scattered moderate showers (10–25 mm) over western Guangdong. By the 6th, a convergence axis intensified over inland hills, producing localized cores up to 50 mm (sky blue to royal blue shading) northeast of Guangzhou.
**Intensification (Sep 7–8):** On September 7, rainfall expanded eastward, with cores exceeding 100 mm over the Pearl River Delta. The 8th marked the peak growth phase: a broad swath of hot pink (100–250 mm) stretched from central Shenzhen northward, triggering flash-flood warnings along tributaries.
**Peak (Sep 9):** September 9 saw the maximum daily accumulation, with dark red (>250 mm) cells persisting near coastal hills. Flood impacts were reported in suburban Foshan and low-lying areas of Dongguan, where drainage systems were overwhelmed.
**Weakening (Sep 10):** By the final day, precipitation waned and shifted southward, contracting to 25–50 mm bands around Hong Kong's northern New Territories. The event concluded with isolated showers and rapid clearing.
**Conclusion**
This multi-day event demonstrated a classic inland-to-coastal propagation of heavy rainfall under synoptic forcing, with peak intensities exceeding 250 mm in localized cores. The spatial shift of maxima and associated flooding underscores the importance of high-resolution reanalysis for regional hazard assessment.

**Example: final_report.md for Tropical Cyclone Event**

**Tropical Cyclone Meteorological Fields (SID: 2022264N17132)**

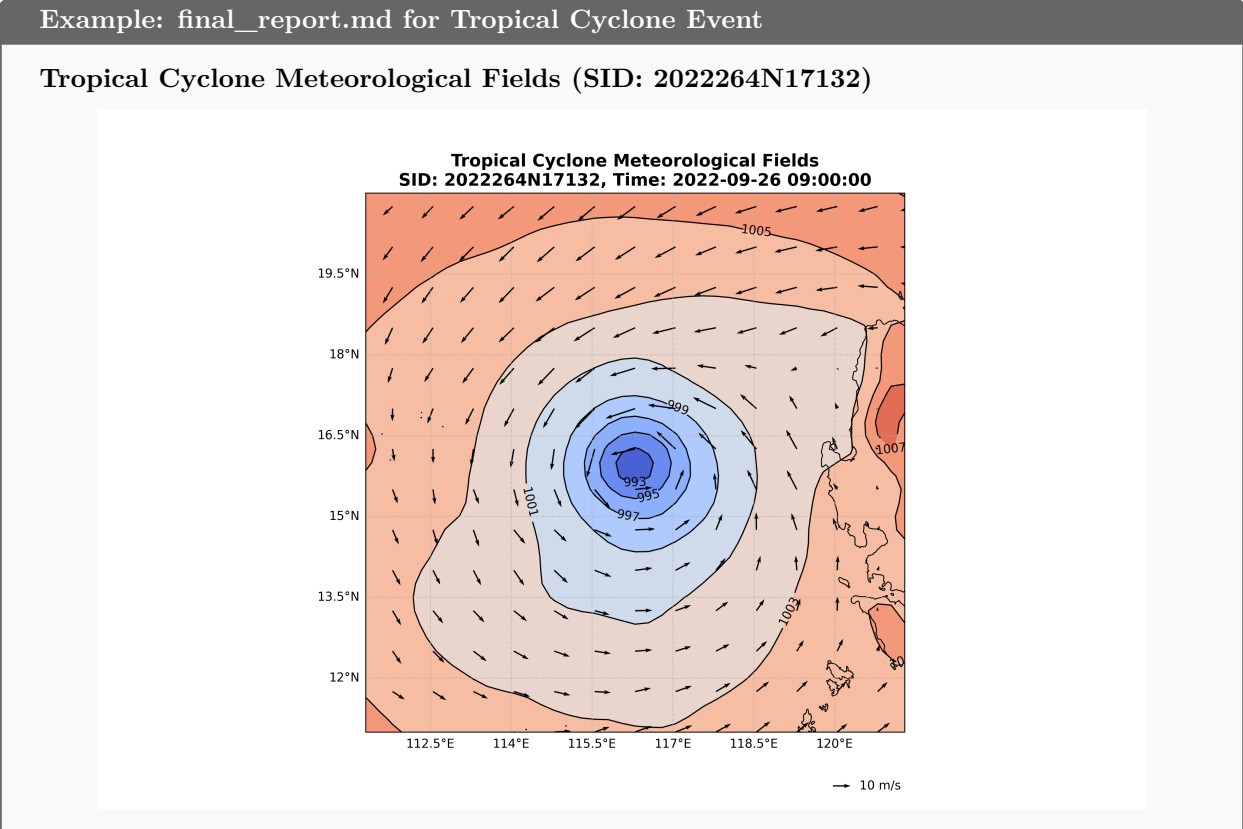

This chart shows a low-pressure system, with a central pressure of approximately 991.8969 hPa, accompanied by a distinct counterclockwise rotating wind field. Such a cyclone may bring strong winds and heavy rainfall, requiring close monitoring of its path and intensity changes to prevent potential impacts on coastal areas or maritime activities.

## B  ClimateAgent Implementation & Reproducibility

This appendix clarifies the concrete software realization of *ClimateAgent* and provides reproducibility details requested by reviewers. In our `app/report-generate/new-approach` implementation, ClimateAgent is a **single-process Python orchestration system** that coordinates modular agent classes (planning, data acquisition, programming, visualization). The orchestrator executes a fixed, plan-driven workflow and persistently stores run state (`context.json`), intermediate artifacts, and logs to a per-run workspace under `new-approach/experiments/`.

### B.1  System Architecture and Entry Points

**Single-process orchestrator.**  ClimateAgent is implemented as a single Python process whose controller is `OrchestratingAgent` (`agents/orchestrating.py`). For each run, the orchestrator (i) creates a task workspace directory `task_YYYYMMDD_HHMMSS` under `new-approach/experiments/` (or resumes an existing one), (ii) runs `PlanningAgent` to obtain a structured subtask list, (iii) routes each subtask to a specialized agent implementation, (iv) executes generated scripts as subprocesses, (v) applies bounded per-subtask retries, and (vi) snapshots the persistent workflow state to `context.json` after planning and after each subtask.

**Modular agents as executable components.**  Each specialized agent is implemented as a Python class that inherits `BaseAgent` and returns a uniform result dictionary (e.g., `status`, `code`, `code_path`, `result`,

error). Agents are instantiated on-demand inside the orchestrator and invoked sequentially as the workflow progresses.

**Runnable interface.** A minimal runnable entry point is provided via `main.py`. It accepts a natural-language task prompt from a text file (`-prompt`) and supports resumption and ablations via `-task_dir`, `-start_from_idx` (0-based), and `-no-context` (disable passing prior context into LLM prompts).

## B.2 Runtime Scheduling, Routing, and Retry Policy

**Plan-driven scheduling.** The orchestrator calls `PlanningAgent` to generate a *plan* as a JSON list of subtasks. Each subtask is a dictionary with minimal schema:

- `subtask`: a short title;
- `agent`: a routing label;
- `description`: a natural-language instruction for downstream execution.

**Agent routing.** Routing is implemented as a fixed mapping from the `agent` string label to an executable agent class:

- `data_download_agent` → `ECMWFDownloadAgent`
- `cdsapi_download_agent` → `CDSAPIDownloadAgent`
- `programming_agent` → `LLMProgrammingAgent`
- `visualization_agent` → `LLMVisualizationAgent`

**Bounded retries at the subtask level.** For each subtask, the orchestrator applies a bounded retry loop (default `max_retrial=3`). Failures are detected by (i) non-zero subprocess return codes when executing generated code or (ii) agent-returned `status="error"`. After exhausting retries, the orchestrator records a failure message in `context.json` (Appendix B.4) and stops the workflow.

**Execution of generated scripts.** Generated code is executed by writing it to a temporary script file `_tmp_script.py` inside the task workspace and invoking `subprocess.run` with:

- **working directory**: the task workspace (`cwd=task_dir`);
- **I/O capture**: `capture_output=True` (`text=True`);
- **timeout**: 6000 seconds per script;
- **interpreter**: invoked as `"python"` (resolved via the system `PATH`).

This design allows the orchestrator to capture stdout/stderr and feed error traces back into subsequent attempts.

## B.3 What We Mean by "Agent": Components vs. Role Prompting

A reviewer concern is whether ClimateAgent is merely a single LLM "role-playing" multiple roles. Our implementation is a **modular agentic workflow** coordinated by an orchestration layer:

- Each "agent" is an **executable software component** (Python class) with a role-specific prompt, an explicit I/O contract (the `BaseAgent` protocol), and dedicated procedures (e.g., metadata scraping, candidate generation, debugging loops).
- The orchestrator invokes agents as separate calls at different workflow stages. For the `programming_agent` and `visualization_agent`, context passing (directory tree, prior codes, README summaries) can be disabled with `-no-context` for ablation.

**Persistence and "memory."** We do not rely on implicit conversational memory as the primary state. Instead, persistence is implemented via:

- a serialized workflow context file (`context.json`);

- on-disk artifacts in the task workspace (downloaded data, generated code, figures, reports, logs).

This externalized state is inspectable, reproducible, and supports explicit resume/recovery (Appendix B.4).

### B.4 Persistent Workflow Context (`context.json`)

**Top-level schema.** In the current `new-approach/experiments` implementation, `context.json` contains four core keys plus one optional key:

- `main_task`: string;

- `subtasks`: list of subtask dictionaries (`subtask`, `agent`, `description`);

- `codes`: mapping from 0-based subtask index (string) to a selected code file path (typically absolute);

- `results`: mapping from 0-based subtask index (string) to captured stdout (success) or a failure message;

- `user_data_summary` (optional): dictionary produced by `UserDataInspectAgent.inspect()`, including paths to `code_output/user_data_summary.md` and `.json`.

The optional `user_data_summary` is included for fresh runs where the orchestrator performs the user-data inspection step; it may be absent for runs started from a user-specified `-task_dir` without prior context.

**Update policy and failure recording.** The orchestrator writes `context.json` (i) after plan generation and (ii) after each subtask finishes (success or failure). Failures are recorded as strings in `results[idx]` (e.g., `"Failed after 3 attempts..."`), rather than in a dedicated `errors` field.

**Resume mechanism.** Resuming is supported by loading `context.json` from an existing `-task_dir`. To continue from an intermediate point, the user specifies `-start_from_idx` (0-based); the orchestrator skips subtasks with indices lower than this value and continues execution from the specified subtask.

**Representative (sanitized) snippet.**

```
{
  "main_task": "Produce the AR ... 100°W-140°W ...",
  "subtasks": [
    {
      "subtask": "Download S2S forecast data for humidity and winds",
      "agent": "data_download_agent",
      "description": "Use the ECMWF S2S dataset ... 2022-03-24 ..."
    }
  ],
  "codes": {
    "0": ".../experiments/task_YYYYMMDD_HHMMSS/data/download_script_attempt_5.py"
  },
  "results": {
    "0": "Submitting retrieval request: {... 'origin': 'ecmf', ...}"
  },
  "user_data_summary": {
```

```
    "markdown": ".../experiments/task_YYYYMMDD_HHMMSS/code_output/user_data_summary.md",
    "json": ".../experiments/task_YYYYMMDD_HHMMSS/code_output/user_data_summary.json",
    "summary": {}
  }
}
```

### B.5 Dynamic API Introspection in Data Acquisition Agents

Valid request parameters (variables, levels, lead times, dataset-specific fields) are constrained and may evolve. Our data acquisition agents therefore include **dynamic API introspection** to ground request generation on runtime metadata, rather than relying on the LLM to guess parameters.

**ECMWF S2S (`ECMWFDownloadAgent`).** The ECMWF S2S download pipeline proceeds as follows:

1. **Parameter extraction**: an LLM parses the user instruction into structured JSON specifying `{parameter, origin, type, mode, date}`. In the current code, parameter extraction uses a stronger model (`gpt-4.1`).

2. **Runtime metadata extraction**: the agent uses Selenium-driven scraping (via `data_download_agent/extract_ecmwf_metadata.py`) to extract valid options (e.g., steps, available levels, and available model-version dates for hindcasts) from the ECMWF dataset pages. Metadata are saved under the task's `data/` directory as JSON (e.g., `meta_{origin}_{param}_{type}_{mode}.json` and, for hindcasts, `hindcast_dates_{origin}_{param}_{type}_{mode}_{selected_model_version_date}.json`).

3. **Multi-candidate script generation**: conditioned on the extracted metadata, the agent generates $n = 8$ candidate download scripts in one LLM call (default model string `o4-mini`).

4. **Validation and execution**: candidates are syntax-checked via `compile(...)`; the orchestrator executes candidates sequentially and accepts the first script with `returncode==0`.

**CDS (`CDSAPIDownloadAgent`).** The CDS download pipeline is:

1. **Dataset selection**: an LLM selects a dataset name from a curated list (`agents/cds_datasets.json`).

2. **Runtime metadata extraction**: the agent uses Selenium to open the dataset download page and parse the form configuration from `__NEXT_DATA__`, saving a JSON metadata file under `<output_dir>/data/<dataset>_metadata.json` (with `output_dir` set to `<task_dir>/data` by the orchestrator).

3. **Multi-candidate script generation**: the current implementation generates $n = 4$ candidate scripts per subtask, conditioned on the metadata, and syntax-checks them with `compile(...)`.

4. **Execution**: the agent can execute valid candidates concurrently (threads + subprocess; 600-second timeout per candidate) and selects the first successful run. Candidate scripts are instructed to print a JSON array of downloaded file paths on the last stdout line for robust parsing and cleanup.

### B.6 Execution Artifacts and Traces

**Workspace artifacts.** Each run creates a task workspace (specified by `-task_dir` or auto-generated under `experiments/`). The workspace contains:

- `context.json`: persistent workflow state (Appendix B.4);

- `data/`: downloaded data, scraped metadata JSONs, and auto-generated download scripts;

- `code/`: generated analysis/visualization scripts;

- `code_output/`: analysis outputs and figures, `user_data_summary.md/json`, and the final report (typically `final_report.md`);

- `log/`: orchestrator and agent logs (e.g., `orchestrating_agent.log`, `programming_agent.log`, `visualization.log`).

**LLM call traces.** LLM call metadata and token usage are recorded in a single JSON file `code_output/token_usage.json`. It stores an `events` list (one entry per API call, including agent name, subtask index, attempt counters, stage labels, model identifier, token counts, and duration) plus a `summary` section aggregating totals and breakdowns by agent/subtask/stage.

### B.7 Benchmark–System Interface

ClimateAgent is a reusable system that accepts natural-language climate workflow prompts as input (e.g., data acquisition, analysis, visualization, and report generation). Benchmark tasks (e.g., prompt collections) are executed by invoking the same runner; evaluation is implemented as a separate harness under `eval_scripts/` (e.g., `evaluate_reports.py` reads `evaluation_pair.json` and compares system vs. reference reports).

**Example commands.**

- Fresh run:

  ```
  python main.py --prompt path/to/task_prompt.txt
  ```

- Resume from an existing workspace starting at subtask $k$:

  ```
  python main.py --prompt path/to/task_prompt.txt \
    --task_dir path/to/task_YYYYMMDD_HHMMSS --start_from_idx k
  ```

- Ablation run without passing context into LLM prompts:

  ```
  python main.py --prompt path/to/task_prompt.txt --no-context
  ```

### B.8 Autonomy Assumptions and Practical Deployment Notes

**Autonomy definition.** By "end-to-end autonomy," we mean that once a task prompt is provided and the runtime environment is configured, the system proceeds without human intervention: it plans, downloads data, generates and executes code, produces visualizations, and writes a report, using bounded retries and error-driven regeneration.

**Required pre-configuration.** Autonomous execution assumes:

- LLM access credentials available via environment variables (the code reads `OPENAI_API_KEY`);

- CDS/ECMWF credentials configured locally (e.g., `cdsapi` and `ecmwf-api-client` credential files);

- Selenium-capable browser tooling available for metadata scraping (headless Chrome via Selenium WebDriver);

- required Python packages installed and network access enabled.

**Failure handling and recovery.** Robustness is achieved by:

- bounded per-subtask retries (`max_retrial`);

- multi-candidate generation for download/code/report steps (defaults: ECMWF $n = 8$, CDS $n = 4$, programming/visualization $n = 4$), with syntax checks and error-driven regeneration;

- runtime error capture via subprocess stdout/stderr, propagated to subsequent attempts;

- persistent state snapshots (`context.json`) enabling resume via `-task_dir` and `-start_from_idx`.

**Security and isolation.** The current implementation does not sandbox generated code beyond subprocess execution. Scripts run with the permissions of the local user environment, using the Python interpreter resolved from `PATH`. We recommend safer deployment practices (e.g., containerization, least-privilege credentials, and network restrictions) for production use.

**Model configuration.** Default model identifiers are configured per agent in code: `PlanningAgent`, `LLMProgrammingAgent`, and `LLMVisualizationAgent` default to `gpt-5`; download-code generation defaults to `o4-mini`; and ECMWF parameter extraction uses `gpt-4.1`. All recorded model identifiers and token usage statistics are saved in `code_output/token_usage.json` for reproducibility.

## C    Executable Examples and Traces.

**Task requirements (`task_SST_1`).** This task analyzes monthly mean sea surface temperature (SST) and SST anomalies in the tropical Pacific during the mature phase of the 2002–2003 El Niño using NOAA OISST v2.1 (AVHRR-only) daily NetCDF files for 2003-01-01–2003-01-31. The required workflow is: (i) download all daily files into `data/` (skip if present; warn and continue on failures); (ii) process local files on the native OISST grid to compute monthly means of `sst` and `anom` over 30°S–30°N and 120°E–290°E (wrapping longitudes to 0–360° as needed); (iii) save processed outputs as NetCDF; (iv) create a publication-quality two-panel map (SST and anomaly) using Cartopy PlateCarree with `central_longitude=-155`, specified contour levels/colormaps, land/coastlines, labeled gridlines, and bold equator and 180° meridian; (v) write a 250–400 word report embedding the figure and documenting data source, methods, and time window.

**Agentic decomposition and responsibilities.** The orchestrator (`orchestrating_agent`) decomposes the task into four subtasks and delegates them to specialized agents:

1. **S1 Download (programming_agent):** loop over 2003-01-01..2003-01-31, construct NOAA URLs, download into `data/`, skip existing, continue on failures.

2. **S2 Process (programming_agent):** open `data/oisst-avhrr-v02r01.200301*.nc`, select surface layer, wrap longitudes if needed, subset to the required domain, compute monthly means, write two NetCDF files.

3. **S3 Plot (programming_agent):** load monthly means, generate two-panel Cartopy figure with required styling, validate PNG output.

4. **S4 Report (visualization_agent):** generate `code_output/final_report.md` embedding the PNG and update `code_output/README.md`.

A lightweight pre-flight step by `user_data_inspect_agent` records a data summary artifact (not used by the OISST pipeline itself).

| Subtask | Agent | Artifacts | Attempts, failures, and fix |
|---|---|---|---|
| S1 Down-load | `programming_agent` | 31 daily NetCDF files in `data/` | 4 code candidates were generated; 2 were rejected due to `SyntaxError: unterminated string literal`. The first executed download script timed out (120 s). A debug regeneration fixed this by adding explicit per-request timeouts and streaming downloads; it then completed and validated the expected 31 files. |
| S2 Process | `programming_agent` | `jan2003_sst_monthly_ mean.nc`, `jan2003_anom_monthly_ mean.nc` | Succeeded on first execution. The pipeline wraps longitudes to 0–360° when needed, subsets to $[-30, 30]°$ latitude and $[120, 290]°$ longitude, computes the mean over `time`, writes outputs, and validates variable/dimension presence. |
| S3 Plot | `programming_agent` | `jan2003_sst_ anom_two_panel.png` | Initial plotting failed with `TypeError: Input z must be 2D, not 3D` (extra singleton dimension). Two debug regenerations later, the fix was to `squeeze(drop=True)` the loaded fields and print dims/shapes before contouring; the figure was then generated and validated. |
| S4 Report | `visualization_ agent` | `final_report.md` | Two attempts failed with `FileNotFoundError: Output directory not found: code_output`. The final fix added an `ensure_directory()` step and successfully wrote the report and updated the README. |

Table 4: SST_1 trace summary (sources under `task_SST_1/log/`.

**Executable example (end-to-end).**

```
cd task_SST_1
python3 code/subtask_1_c896079c_cand0.py
python3 code/subtask_2_18510429_cand0.py
python3 code/subtask_3_076f29d8_cand0.py
python3 code/subtask_4_636361f2_viz_cand0_debug2.py
```

Expected trace highlights (selected stdout lines saved in logs):

```
[INFO] Saved SST monthly mean to code_output/jan2003_sst_monthly_mean.nc
[INFO] Saved anomaly monthly mean to code_output/jan2003_anom_monthly_mean.nc
[DEBUG] SST dims: ('lat', 'lon'), shape: (240, 680)
[INFO] Figure saved to code_output/jan2003_sst_anom_two_panel.png
```

**Trace overview (attempts, failures, fixes).**

**Selected raw failure traces (minimal excerpts). S1 generation/runtime issues** (`task_SST_1/log/programming_agent.log`):

```
Syntax error: unterminated string literal (subtask_1_c896079c_cand1.py)
... timed out after 120 seconds (subtask_1_c896079c_cand0.py)
```

**S3 dimensionality error** (`task_SST_1/log/43a886fb.log`):

```
TypeError: Input z must be 2D, not 3D
```

**S4 missing output directory** (`task_SST_1/log/visualization.log`):

```
FileNotFoundError: Output directory not found: code_output
```

**Final artifacts.** The run produces the following deliverables under `task_SST_1/code_output/`:

- `jan2003_sst_monthly_mean.nc` (monthly mean absolute SST), `jan2003_anom_monthly_mean.nc` (monthly mean SST anomaly);

- `jan2003_sst_anom_two_panel.png` (two-panel map; 3000×3600, 300 DPI);

- `final_report.md` (250–400 word summary report embedding the figure).

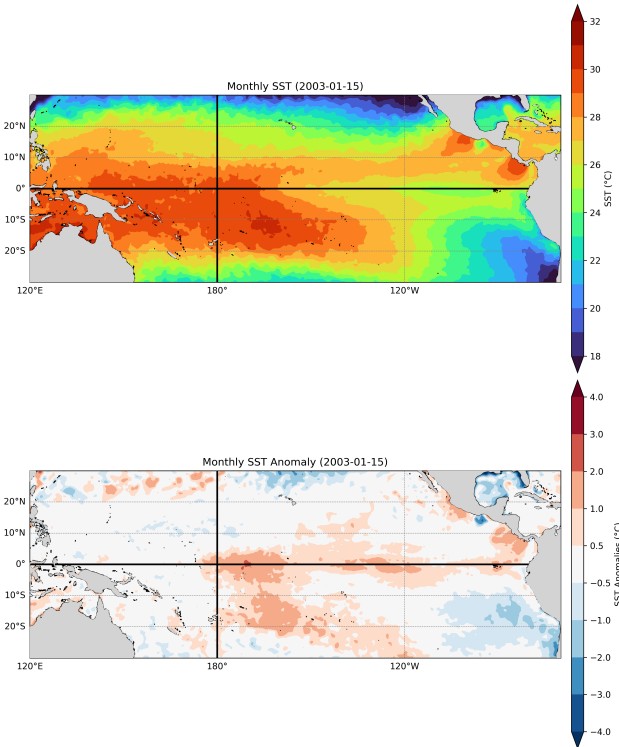

Figure 7: Monthly mean SST (top) and SST anomaly (bottom) for January 2003 from NOAA OISST v2.1 (AVHRR-only), averaged over 2003-01-01–2003-01-31 and plotted over 30°S–30°N, 120°E–290°E.

**Final result report (generated text).** The figure(Figure 7) above presents a two-panel map of monthly mean sea surface temperature (SST) and SST anomalies for January 2003 over the tropical Pacific (30°S–30°N, 120°E–290°E), centered on 2003-01-15. In the upper panel, absolute SST values range from about 18°C in the eastern basin to over 30°C in the western warm pool. The highest temperatures ($>28$°C) form a continuous eastward extension along the equator, peaking near 160°W. Cooler waters ($<24$°C) occupy the eastern Pacific near South America and the subtropical gyres north and south of the equator.

The lower panel shows SST anomalies relative to the 1971–2000 climatology. A pronounced El Niño signature emerges: positive anomalies up to $+3$°C span from the dateline ($\sim 180$°E) eastward to around 240°E, concentrated within $\pm 5$° of the equator. The warm anomaly tongue extends approximately from 140°E to 260°E. Opposing negative anomalies ($\sim -0.5$°C to $-1$°C) appear in the western margin of the tropical Pacific and in higher-latitude subtropical regions, indicating compensating cooling outside the core equatorial zone. Overall, anomalies exceed $+2$°C across a broad central–eastern band, confirming the mature phase of the 2002–2003 El Niño.

Data were sourced from NOAA OISST v2.1 AVHRR-only daily NetCDF files for January 1–31, 2003. Files were downloaded locally and processed using xarray: surface fields of `sst` and `anom` were combined and averaged over the month. Subsetting used native longitudes (0–360°) re-wrapped to 120°E–290°E and latitudes $-30$° to $+30$°.

Visualization employed matplotlib and cartopy (PlateCarree projection with `central_longitude=-155`), using a sequential colormap for SST (18°C–32°C, 1°C intervals) and a diverging colormap for anomalies (levels $-4$°C to $+4$°C).

## D  Component-level Ablation Analysis

The reviewer requests component-level evidence that attributes performance gains to specific parts of CLIMATEAGENT. Because CLIMATEAGENT is an end-to-end executable workflow, some components (notably planning and data acquisition) are *necessary for executability*: removing them entirely would collapse the pipeline and does not yield a meaningful runnable baseline. Therefore, in this section we provide (i) **direct ablations where the system remains executable** (e.g., Contextual Coordination on/off; Sec. D.2), and (ii) **trace-based component attribution** for Coordinated Task Planning (CTP), using concrete plans, artifact chains, and per-subtask attempt logs to isolate the role CTP plays in long-horizon completion and error localization.

### D.1  Ablation Analysis on Coordinated Task Planning (CTP)

**What CTP does in our implementation.** Coordinated Task Planning (CTP) is the orchestration mechanism that transforms a high-level scientific request into an ordered list of subtasks, assigns each subtask to a capability-matched agent (download vs. programming vs. visualization), and enforces explicit artifact hand-offs (filenames/paths and expected formats) between subtasks. In our system, the resulting subtask plan is persisted in `context.json`, and execution is driven by a deterministic loop over these subtasks with per-subtask retries recorded in `log/orchestrating_agent.log` (see Appendix B.4–B.6 for implementation details and traces).

**Why we use trace-based attribution for CTP.** CTP cannot be "turned off" without making the workflow non-executable (e.g., there is no downstream routing without a subtask list). Instead of a non-runnable ablation, we attribute CTP's contribution via **observable execution evidence**:

- **Plan validity:** whether the plan covers the essential scientific stages (data acquisition $\rightarrow$ computation $\rightarrow$ visualization/reporting) in an expert-reasonable order.

- **Artifact-chain explicitness:** whether intermediate outputs are explicitly named and reused (e.g., GRIB/NetCDF intermediates feeding downstream steps), enabling reproducibility.

- **Failure localization:** whether failures are contained to a single subtask and resolved via local retries without discarding previously completed stages.

- **Capability boundary compliance:** whether tasks that require separation of responsibilities (e.g., generating a downloader vs. executing downloads into `data/`) are handled via explicit hand-offs.

**Representative long tasks.** To ground the above signals, we report three representative long-horizon tasks (6–7 subtasks) that require multi-stage coordination and produce complete artifact chains: `task_AR_2` (AR detection + weekly frequency map, US West Coast), `task_AR_14` (same workflow for East Asia; includes permission-aware download separation), and `task_DR_6` (SPI-based drought severity mapping over CONUS). Table 5 summarizes their subtask counts, total attempts (including retries), agent switches, and wall-clock times, derived from `context.json` and orchestrator logs.

**Evidence from the traces.** Across these tasks, CTP produces plans that (i) explicitly separate data acquisition, computation, and reporting, and (ii) establish a concrete artifact chain that enables downstream stages to consume intermediate outputs deterministically. Moreover, when errors occur, the orchestrator retries the failing subtask while preserving already-produced artifacts, demonstrating localized recovery rather than restarting the entire workflow. Below we summarize the CTP evidence for each representative task.

| Task | #Subtasks | #Attempts | Agent switches | Wall-clock |
|---|---|---|---|---|
| task_AR_2 | 6 | 6 | 3 | 27m34s |
| task_AR_14 | 7 | 8 | 3 | 47m52s |
| task_DR_6 | 7 | 7 | 2 | 26m21s |

Table 5: CTP trace summary for three representative long-horizon tasks (from `*/context.json` and `*/log/orchestrating_agent.log`).

### Example 1: `task_AR_2` (AR detection + weekly frequency map; 6 subtasks)

**Plan validity and artifact chain.** The plan explicitly separates setup/config, data acquisition, scientific computation, and reporting, with named hand-offs: GRIB downloads → daily IVT NetCDFs → daily AR presence/pathway outputs → weekly aggregation → map PNG → report. These artifacts are written under a structured directory (`code_output/outputs_ar_freq/`) and referenced by subsequent subtasks, supporting reproducibility and deterministic execution.

**Failure localization.** When a downstream stage fails, the orchestrator retries only that subtask (recorded in `orchestrating_agent.log`) and reuses previously generated intermediates rather than restarting the pipeline, illustrating the intended CTP behavior for long-horizon robustness.

### Example 2: `task_AR_14` (AR detection + weekly frequency map; 7 subtasks)

**Capability boundary compliance.** This task requires a coordination-critical split: a programming agent *generates* a downloader script, while a dedicated download agent *executes* it to write into `data/`. CTP makes this hand-off explicit in the plan (separate subtasks with explicit artifacts), enabling a policy-compliant, reproducible workflow in which the download script is checked into the workspace and the data-write step is isolated.

**Failure localization.** The orchestrator retries a failing scientific subtask without invalidating the completed download and intermediate products, illustrating localized recovery in a multi-stage artifact chain.

### Example 3: `task_DR_6` (SPI drought severity mapping over CONUS; 7 subtasks)

**Plan validity aligned with domain workflow.** The drought pipeline is decomposed into statistically meaningful stages (download → crop → climatology split → monthly stats → SPI → map → report). This decomposition reduces the risk of silent methodological errors (e.g., mixing climatology and target windows) by making intermediate stages explicit and inspectable.

**Artifact-chain explicitness and localized retries.** Intermediate products (cropped dataset, climatology statistics, SPI field) are saved with explicit filenames and consumed downstream. When failures occur, retries target a single stage, preserving earlier artifacts and preventing unnecessary recomputation.

**Takeaway.** These traces provide component-level attribution for CTP: it yields expert-reasonable decompositions, explicit artifact chains, capability-aware routing, and localized recovery behavior in long-horizon climate workflows. This complements the direct on/off ablation for Contextual Coordination in Sec. D.2, which quantifies quality drops when cross-step context alignment is removed.

### D.2 Ablation Analysis on Contextual Coordination

Contextual Coordination is the mechanism that maintains cross-step consistency by (i) propagating structured task state (e.g., key file paths, intermediate outputs, and constraints) between subtasks and agents, and (ii) enforcing that downstream steps consume the *actual* artifacts produced upstream rather than re-generating or guessing them. This is particularly important for long-horizon workflows where subtle context drift (e.g., mismatched filenames, inconsistent regions/time windows, or re-derived parameters) can silently degrade scientific correctness or cause late-stage failures.

| Task ID | Category | With Contextual Coordination | | | | | w/o Contextual Coordination | | | | |
|---------|----------|-------|------|-------|------|--------|-------|------|-------|------|--------|
| | | Read. | Sci. | Comp. | Vis. | Report | Read. | Sci. | Comp. | Vis. | Report |
| AR1 | AR | 7 | 6 | 5 | 8 | 6.5 | 5 | 3 | 4 | 6 | 4.5 |
| DR1 | DR | 8 | 9 | 7 | 8 | 8.0 | 7 | 8 | 6 | 7 | 7.0 |
| EP1 | EP | 8 | 9 | 7 | 9 | 8.25 | 7 | 8 | 6 | 5 | 6.5 |
| HW1 | HW | 9 | 8 | 9 | 8 | 8.5 | 9 | 8 | 9 | 7 | 8.3 |
| SST1 | SST | 9 | 10 | 9 | 8 | 9.0 | 6 | 5 | 4 | 7 | 5.5 |
| TC1 | TC | 9 | 8 | 9 | 8 | 8.5 | err | err | err | err | err |

Table 6: Ablation on Contextual Coordination. Read.=Readability, Sci.=Scientific Rigor, Comp.=Completeness, Vis.=Visual Quality, Report=overall report score. "err" indicates the run failed to produce a valid end-to-end output.

**Ablation setup.** We compare two settings on six representative tasks (one per domain):

- **With Contextual Coordination:** the orchestrator passes forward the persisted workflow state and key artifact pointers, enabling agents to reference previously produced outputs and maintain consistent assumptions across subtasks.

- **w/o Contextual Coordination:** we remove this cross-step context propagation, forcing each subtask to proceed with limited knowledge of prior artifacts and decisions, which increases the risk of drift and brittle interfaces.

**Evaluation.** We evaluate report quality using the same rubric dimensions as in the main paper: Readability, Scientific Rigor, Completeness, Visual Quality, and the aggregated Report Score. We additionally record whether the run produces a valid final output chain (required figure + report). Table 6 reports results.

**Results and attribution.** Removing Contextual Coordination causes consistent degradations in report quality across domains, with particularly large drops on tasks that require careful cross-step reuse of intermediate scientific artifacts.

- **Large quality drops on artifact-heavy workflows.** SST1 shows a substantial decline in overall Report Score (9.0 → 5.5), accompanied by sharp reductions in Scientific Rigor and Completeness. This is consistent with context drift causing downstream steps to mis-handle intermediate datasets, variable choices, or time/region settings when prior decisions are not propagated.

- **Cross-step drift affects scientific correctness signals.** In AR1 and EP1, the largest decreases occur in Scientific Rigor / Completeness and Visual Quality, reflecting that downstream code and plots are more likely to deviate from upstream constraints (e.g., required thresholds, plotting levels, or spatial aggregation definitions) when the system cannot reliably reference the correct intermediate artifacts.

- **Hard failure in long-horizon pipelines.** TC1 fails entirely ("err") without contextual coordination, indicating that brittle multi-step dependencies (e.g., intermediate text/figures written in earlier subtasks) can break at the final stage when artifact paths and working-directory assumptions are not consistently maintained.

**Takeaway.** This ablation provides direct evidence that Contextual Coordination is a key contributor to end-to-end robustness and report quality in CLIMATEAGENT. By maintaining consistent state and artifact references across agents and subtasks, it reduces context drift, prevents late-stage contract breaks, and improves scientific completeness and rigor in the generated reports.

Table 7: Task stats and errors (short cols)

| Name | Subtask | ProgRet | DataRet | VizRet | ReqErr | ShpKey | MiscErr | SynErr | Timeout | TypeErr |
|---|---|---|---|---|---|---|---|---|---|---|
| task_AR_1 | 4 | 3 | 2 | 0 | 1 | 2 | 1 | 0 | 0 | 0 |
| task_AR_2 | 6 | 0 | 0 | 0 | 0 | 0 | 0 | 0 | 0 | 0 |
| task_AR_3 | 5 | 2 | 0 | 4 | 0 | 1 | 4 | 4 | 1 | 1 |
| task_AR_4 | 7 | 8 | 0 | 0 | 0 | 0 | 2 | 3 | 0 | 8 |
| task_AR_5 | 5 | 3 | 0 | 1 | 0 | 2 | 4 | 4 | 0 | 4 |
| task_DR_1 | 4 | 0 | 1 | 0 | 1 | 0 | 0 | 0 | 0 | 0 |
| task_DR_2 | 4 | 1 | 0 | 7 | 0 | 0 | 7 | 1 | 0 | 0 |
| task_DR_3 | 4 | 3 | 0 | 6 | 0 | 0 | 6 | 1 | 0 | 2 |
| task_DR_4 | 4 | 1 | 11 | 1 | 11 | 0 | 2 | 0 | 0 | 0 |
| task_DR_5 | 5 | 1 | 0 | 0 | 0 | 0 | 1 | 0 | 0 | 0 |
| task_EP_1 | 4 | 2 | 0 | 2 | 0 | 2 | 2 | 2 | 0 | 0 |
| task_EP_2 | 4 | 2 | 0 | 2 | 0 | 2 | 1 | 1 | 0 | 0 |
| task_EP_3 | 5 | 4 | 0 | 1 | 0 | 3 | 2 | 0 | 0 | 0 |
| task_EP_4 | 4 | 3 | 1 | 1 | 0 | 2 | 1 | 4 | 0 | 1 |
| task_EP_5 | 4 | 1 | 0 | 43 | 0 | 1 | 22 | 20 | 0 | 0 |
| task_HW_1 | 2 | 0 | 0 | 0 | 0 | 0 | 0 | 0 | 0 | 0 |
| task_HW_2 | 2 | 0 | 0 | 0 | 0 | 0 | 0 | 0 | 0 | 0 |
| task_HW_3 | 2 | 0 | 0 | 0 | 0 | 0 | 0 | 0 | 0 | 0 |
| task_HW_4 | 2 | 0 | 0 | 0 | 0 | 0 | 0 | 0 | 0 | 0 |
| task_HW_5 | 2 | 1 | 0 | 0 | 0 | 0 | 0 | 0 | 0 | 1 |
| task_SST_1 | 4 | 3 | 0 | 2 | 0 | 0 | 2 | 2 | 1 | 2 |
| task_SST_2 | 4 | 10 | 0 | 0 | 1 | 1 | 3 | 3 | 2 | 0 |
| task_SST_3 | 4 | 3 | 0 | 0 | 0 | 1 | 0 | 2 | 0 | 0 |
| task_SST_4 | 4 | 1 | 0 | 1 | 0 | 0 | 2 | 0 | 0 | 0 |
| task_SST_5 | 4 | 6 | 0 | 0 | 0 | 1 | 0 | 5 | 0 | 0 |

### D.3 Ablation Analysis on Adaptive Self-Correction

Adaptive Self-Correction (ASC) is the mechanism that improves CLIMATEAGENT robustness by iteratively recovering from execution-time failures. In our implementation, ASC operates through bounded retry-and-regenerate loops: when a subtask fails, the responsible agent (programming/data/visualization) generates revised candidates conditioned on observed error messages and re-executes until success or the retry budget is exhausted. This behavior is essential in long-horizon climate workflows where failures arise from heterogeneous I/O backends, evolving APIs, and brittle code/plotting details.

**Why trace-based attribution (instead of full removal).** In an end-to-end executable pipeline, ASC is tightly coupled with the orchestrator's runtime semantics: subtasks are sequentially dependent, and failures are handled by retries and regeneration. Rather than introducing additional experimental variants in this revision, we provide **trace-based component attribution** grounded in the errors that were *actually encountered* and the recovery events that were *required* for successful completion. Under the orchestrator's fail-stop behavior at the subtask level, any subtask that required at least one recovery would have terminated at its first failure without ASC, thereby preventing downstream subtasks (and thus the final report) from being produced.

**Signals from execution artifacts and logs.** We summarize ASC behavior with two log-derived signals:

- **Retry volume by stage:** the number of retries required for each task in the Programming, Data, and Visualization stages (`ProgRet`, `DataRet`, `VizRet`).

- **Recovered error taxonomy:** counts of observed error classes, including request/API errors (`ReqErr`), shape/key errors (`ShpKey`), miscellaneous runtime errors (`MiscErr`), syntax errors (`SynErr`), timeouts (`Timeout`), and type errors (`TypeErr`).

Table 7 reports these statistics aggregated from execution logs across representative tasks.

**Findings.**    Two consistent patterns emerge from Table 7.

**(1) Recovery is frequently necessary for completion.** Many tasks exhibit non-zero retry counts (e.g., `ProgRet`$> 0$ and/or `VizRet`$> 0$), indicating that the first generated candidate is often insufficient and that successful end-to-end runs depend on iterative correction. Because subtasks are sequentially dependent, a single unrecovered failure would block all downstream stages and prevent producing the final figure and report.

**(2) Failure modes are heterogeneous and stage-specific.** The recovered error taxonomy reveals that different stages fail in different ways:

- **Programming-stage failures** (e.g., `ShpKey`, `TypeErr`, `MiscErr`) reflect common scientific-computing issues such as coordinate/dimension mismatches, variable naming assumptions, or xarray backend/engine mismatches.

- **Visualization-stage failures** (often reflected in elevated `VizRet` and `SynErr`/`MiscErr`) frequently stem from brittle artifact references (paths/filenames) and plotting boilerplate issues; ASC enables iterative fixes without rerunning the entire workflow.

- **Data-stage failures** (captured by `DataRet` and `ReqErr`) are driven by external service instability, request/credential issues, or transient connectivity problems; ASC mitigates these via retries and regenerated download candidates.

**Interpretation and linkage to concrete failures.**    These trace-based statistics attribute a concrete role to ASC: it resolves diverse runtime failures that occur in practice and would otherwise terminate subtasks under fail-stop execution semantics. While final report scores reflect output quality, ASC primarily contributes by improving **robustness and completion** under realistic conditions (external APIs, heterogeneous file formats, and fragile artifact interfaces). We further connect these error classes to concrete examples with logs and intermediate artifacts in Appendix E.

**Takeaway.**    Adaptive Self-Correction is a key robustness component in CLIMATEAGENT, enabling recovery from syntactic, runtime, data I/O, and external-service failures that are common in long-horizon climate workflows and that would otherwise block end-to-end execution.

# E    Dedicated analysis of failure cases

Given the complexity of the multi-agent workflow (data acquisition → scientific computation → visualization/reporting), failures typically arise from (i) external data services, (ii) heterogeneous file formats and coordinate conventions, (iii) runtime/resource constraints, and (iv) brittle artifact contracts (paths/names) across agents. Below we analyze four representative cases of CLIMATEAGENT, including log evidence, intermediate artifacts, underlying causes, and mitigations done by agents.

### E.1    Case 1: External data service failure (connection error during download)

**Where it happens.** Data acquisition stage (download agent), `task_AR_1`.

**Symptom / impact.** The workflow cannot proceed because the required forecast file is missing; the orchestrator retries.

**Log evidence.** The orchestrator records a connection error on the first attempt:

```
task_AR_1/log/orchestrating_agent.log:18-20
Attempt 1 for subtask 1
Subtask 1 failed: Connection error.
Attempt 2 for subtask 1
```

**Intermediate artifacts.**

- Expected (after success): `task_AR_1/data/s2s_IAPCAS_20220324_030.nc`.

- Download implementation: `task_AR_1/data/download_script_attempt_2.py`.

**Underlying cause.** Reliance on an external API (ECMWF Web API) introduces transient network/service failures; the system has no control over availability.

**Mitigation / resolution.**

- **Retry with backoff and idempotency**: retries should include exponential backoff and resume-safe downloads.

- **Preflight credential and endpoint checks**: detect missing credentials or endpoint issues early with a lightweight request.

- **Cache-first behavior**: if the expected file exists and is valid (non-empty, readable), skip re-download.

### E.2 Case 2: Data format and coordinate heterogeneity (xarray engine + concat failures)

**Where it happens.** Scientific computation stage (programming agent), `task_AR_1` subtask 2.

**Symptom / impact.** Multiple candidates fail during dataset loading: (i) `xarray.open_mfdataset` cannot infer concatenation order for threshold files, and (ii) the forecast file cannot be opened using default NetCDF backends (requires GRIB/cfgrib).

**Log evidence.** Threshold concatenation failure:

```
task_AR_1/log/programming_agent.log:851
ValueError: Could not find any dimension coordinates to use to order the
datasets for concatenation
```

Forecast file backend/engine mismatch (default engines fail; cfgrib succeeds):

```
task_AR_1/log/programming_agent.log:10364
Engine 'default' failed: did not find a match in any of xarray's currently
installed IO backends [...]
Engine 'netcdf4' failed: NetCDF: Unknown file format: '...s2s_IAPCAS_...nc'
...
Opened S2S with engine='cfgrib'
```

**Intermediate artifacts.**

- Forecast file: `task_AR_1/data/s2s_IAPCAS_20220324_030.nc`.

- User thresholds directory: `../user_provided_data/predicted_IVT_thres/`.

- Candidate implementations: `task_AR_1/code/subtask_2_*.py`.

**Underlying cause.** "Same extension, different reality": files named `.nc` may be NetCDF or GRIB-derived containers depending on the download pipeline and libraries available; additionally, multi-file threshold datasets may not share coordinates needed for `combine="by_coords"` concatenation.

**Mitigation / resolution.**

- **Robust I/O strategy**: attempt multiple engines; explicitly use `engine="cfgrib"` with `indexpath=""` when GRIB-like.

- **Coordinate standardization**: normalize `time/lat/lon/level` names and ensure `time` is a dimension before slicing.

- **Threshold ingestion that does not assume concat coords**: open files one-by-one, extract/average the threshold variable, and regrid/align explicitly (or fail fast with a clear diagnostic about which coordinate is missing).

- **Validation gates**: before heavy computation, assert non-empty spatial/time subsets and coordinate consistency between forecast and thresholds.

### E.3 Case 3: Runtime constraints (timeouts + invalid code candidates)

**Where it happens.** Early-stage programming tasks with heavy I/O or expensive operations (example: `task_SST_1` subtask 1).

**Symptom / impact.** Some LLM-generated candidates fail immediately due to syntax errors; others time out under the orchestrator's execution limit, causing retries and delayed completion.

**Log evidence.**

```
task_SST_1/log/programming_agent.log:1-4
Syntax error: unterminated string literal (...) (subtask_1_..._cand1.py)
Syntax error: unterminated string literal (...) (subtask_1_..._cand3.py)
Exception during execution (attempt 1): ... timed out after 120 seconds
```

**Intermediate artifacts.**

- Candidate scripts: `task_SST_1/code/subtask_1_c896079c_cand*.py`.

- Partially produced outputs (after eventual success): `task_SST_1/data/` downloads and `task_SST_1/code_output/` NetCDF outputs.

**Underlying cause.** Two distinct issues: (i) generative code may be syntactically invalid, and (ii) even valid code may exceed a fixed wall-clock budget due to large downloads, slow per-file loops, or non-vectorized computations.

**Mitigation / resolution.**

- **Pre-execution lint/syntax gate**: run a fast parser check (`python -m py_compile`) before full execution to avoid wasting the timeout budget.

- **Chunked and resumable I/O**: download in chunks with checkpointing; avoid re-downloading existing files.

- **Performance-aware prompting and templates**: prefer known-efficient patterns (vectorization, xarray operations, bounded spatial windows) and avoid Python-level loops over grids.

- **Adaptive timeouts**: allow longer budgets for known-heavy subtasks, or add progress heartbeats so the orchestrator can distinguish a hang from a slow but healthy job.

### E.4 Case 4: Artifact contract break (path/working-directory mismatch halts final reporting)

**Where it happens.** Reporting stage (visualization agent), `task_TC_1` subtask 10.

**Symptom / impact.** The workflow fails after three attempts at the final reporting step because the visualization agent cannot find an intermediate file that *does exist*, but at a different relative path than assumed.

**Log evidence.** Subtask 9 successfully writes `code_output/central_pressure_...txt`, but subtask 10 searches under `code/code_output` and fails repeatedly:

```
task_TC_1/log/orchestrating_agent.log:2833-2847
Subtask 9 succeeded.
--- Subtask 10/10: Read 'code_output/central_pressure_...txt' ...
Subtask 10 failed with error:
... file not found at ...\code\code_output\central_pressure_2023036S12117.txt
```

**Intermediate artifacts.**

- Produced by subtask 9 (exists): `task_TC_1/code_output/central_pressure_2023036S12117.txt`.

- Produced earlier (exists): `task_TC_1/code_output/meteorological_fields_2023036S12117_20230207_2100.png`.

- Missing due to path mismatch: the report file `task_TC_1/report_2023036S12117.md` is not generated.

**Underlying cause.** A brittle contract between agents regarding the working directory and relative paths. The visualization agent implicitly ran as if its CWD were `task_TC_1/code/`, while the artifact was written relative to the task root.

**Mitigation / resolution.**

- **Enforce a single task-root CWD**: the orchestrator should run all agent scripts from the task root and forbid ad-hoc CWD changes.

- **Resolve paths from an explicit root**: use a declared `TASK_ROOT` and construct paths via `Path(TASK_ROOT)/"code_output"/....`

- **Preflight artifact checks**: before generating reports, validate required inputs exist (and are non-empty) and emit a clear, single error message with the checked paths.

- **Contract tests**: add small automated checks that the expected filenames/locations match the previous subtask outputs.

# F   Human Expert vs. LLM-as-Judge Agreement

To validate the reliability of our LLM-as-a-judge evaluation, we conduct a human expert assessment on the same set of generated reports and compare domain-expert scores with the LLM judge scores. For each climate domain, we compute the mean absolute difference between the human expert score and the LLM judge score (lower is better, indicating stronger agreement).

## F.1   Agreement Results Across Domains

Table 8 summarizes the expert–LLM score gaps across the six domains in CLIMATE-AGENT-BENCH-85.

| Domain | AR | DR | EP | HW | SST | TC |
|---|---|---|---|---|---|---|
| Mean absolute score difference ($|s_{\text{expert}} - s_{\text{LLM}}|$) | 1.9167 | 0.3250 | 0.5500 | 0.4250 | 0.4750 | 0.4750 |

Table 8: Human expert vs. LLM-as-judge agreement by domain. Values are mean absolute score differences; smaller values indicate closer agreement.

Overall, the agreement is high in five out of six domains: **DR, EP, HW, SST, and TC** all exhibit gaps around ∼0.3–0.55, suggesting that the LLM judge largely tracks expert preferences in these settings. In contrast, **AR shows a substantially larger discrepancy** (1.9167). This indicates that the Atmospheric River tasks—often involving stricter visualization requirements and more nuanced spatial diagnostics—can amplify differences between what an expert prioritizes and what an LLM judge infers from the report and figures alone. We therefore perform a targeted qualitative analysis of the AR domain below.

### F.2 Analyzing the Most Complex Task: Atmospheric River Example

We analyze two representative AR cases to better understand the source of disagreement between the expert and the LLM judge. In both cases, we compare the AI Agent output to a Specialist-produced reference.

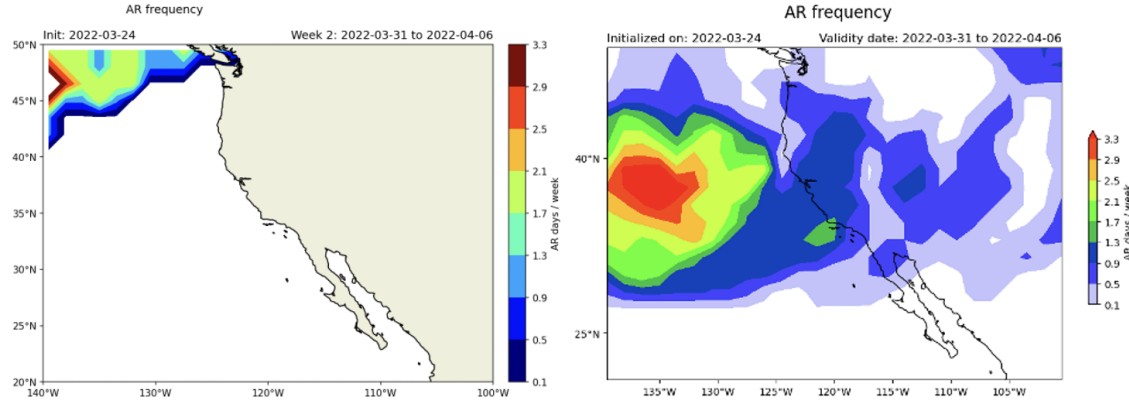

Figure 8: Case 1. Left: Output from the AI Agent. Right: Created by Specialist.

**Case 1.**

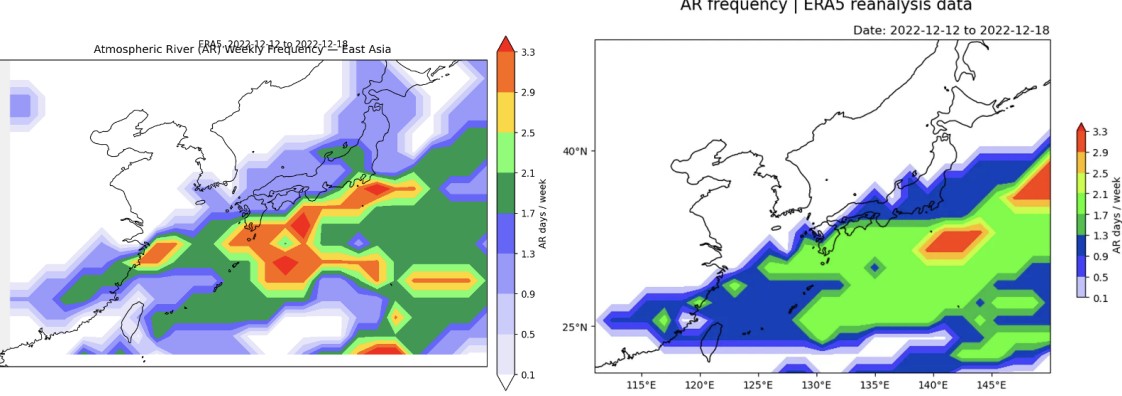

Figure 9: Case 2. Left: Output from the AI Agent. Right: Created by Specialist.

**Case 2.**

**Comments.** Across the two cases, the written organization and narrative framework are broadly similar; the primary difference lies in the visual quality of the AI Agent—Case 1 omits critical information, whereas Case 2 reproduces the main spatial patterns reasonably well.

- **Readability.** In both cases, the text uses a region-by-region, bullet-point format with clear logic and generally appropriate scientific terminology. The descriptions of data values and geographic locations are explicit and accessible to the intended expert audience. However, the phrasing and terminology are relatively repetitive, with limited linguistic variety.

- **Scientific Rigor.** The data sources are clearly stated and credible, and the interpretation of spatial patterns is generally sound, covering the major regional distributions of AR frequency. Nonetheless, neither case discusses forecast uncertainty or potential model biases, yielding a comparable, moderate level of rigor overall.

- **Completeness.** Both narratives cover the standard elements of "region–intensity–hotspots/corridors" and follow a similar structure, but neither includes an assessment of potential impacts or implications.

- **Visual Quality.** In Case 1, the AI Agent figure shows an obvious omission/anomaly (the principal high-frequency band and hotspots are not depicted), which substantially weakens—if not invalidates—the evidential basis for the subsequent interpretation. In Case 2, the AI Agent figure is largely consistent with the Specialist's map (a pronounced AR band and an offshore high-value core are evident); remaining differences are minor in extent, thickness, or position and fall within acceptable plotting variability.

**Takeaway.** These examples suggest that the larger expert–LLM gap in AR is driven primarily by **visual-evidence sensitivity**: domain experts penalize figure omissions or physically implausible spatial patterns more strongly, whereas an LLM judge may overweight the narrative structure when the text appears coherent. This motivates (i) stronger visualization validation checks and (ii) incorporating a small amount of expert auditing for visually critical tasks.

## G  Computation Cost

This section quantifies the computational overhead of ClimateAgent in terms of (i) LLM token usage and (ii) end-to-end wall-clock time. Since ClimateAgent uses multi-candidate generation (e.g., $n=8$ in data agents) and iterative debugging/retries, reporting these costs is important for interpreting robustness gains.

**Per-agent token/time breakdown.** Table 9 reports the breakdown by domain (AR/DR/EP/HW/SST) and by agent type: Planning (Plan), Programming (Prog), and Visualization (Viz). Across the five representative domains, ClimateAgent consumes a total of **664,896 input tokens** and **938,365 output tokens** (**1,603,261 total tokens**), with an aggregated wall-clock time of **8,094,272 ms** ($\approx$ **134.9 min**). Wall-clock time includes end-to-end execution (tool calls, downloads, Python script execution), not only LLM inference.

**Where the overhead comes from.** As expected, the dominant cost is in the Programming agent, which performs iterative code generation and debugging: **Prog accounts for 729,331/938,365 = 77.7% of output tokens** and most of the wall-clock duration across tasks. Planning is comparatively lightweight (one call per task in these runs), while Visualization can incur additional calls when report synthesis or figure embedding needs refinement (e.g., HW).

**Comparison to a single-pass GPT-5 baseline (tokens).** To contextualize the overhead of multi-step agentic execution, Table 10 compares **output tokens** (OtTk) between ClimateAgent and a GPT-5 baseline for Task 1 in each domain. Summed over the five domains, ClimateAgent produces **938,365 output tokens** versus **176,220 output tokens** for the GPT-5 baseline (Task 1), reflecting the additional intermediate reasoning, multi-candidate generation, and iterative debugging in ClimateAgent. We emphasize that this overhead directly supports higher robustness in tool-augmented, long-horizon workflows by reducing the probability of unrecoverable failures.

## H  Comparison to a General-Purpose Multi-Agent Framework (LangGraph DeepAgent)

**Background and motivation.** Our main experiments compare ClimateAgent against strong *generic* baselines (e.g., a single-pass GPT-5 workflow and GitHub Copilot). However, these comparisons do not fully

Table 9: CLIMATEAGENT computational cost and wall-clock time breakdown by domain and agent. Plan/Prog/Viz denote the Planning/Programming/Visualization agents. SubT = number of subtasks assigned to the agent; Invk = number of LLM invocations; InTk/ OtTk = input/output tokens; Dur = wall-clock duration in milliseconds for that agent; TotO = total output tokens summed over Plan+Prog+Viz; TotD = total duration (ms) summed over Plan+Prog+Viz.

| Task | Plan | | | | | Prog | | | | | Viz | | | | | Total | |
| | SubT | Invk | InTk | OtTk | Dur | SubT | Invk | InTk | OtTk | Dur | SubT | Invk | InTk | OtTk | Dur | TotO | TotD |
|---|---|---|---|---|---|---|---|---|---|---|---|---|---|---|---|---|---|
| AR | 1 | 1 | 9630 | 6046 | 103138 | 4 | 15 | 193785 | 233401 | 1810552 | 1 | 1 | 3481 | 25769 | 138872 | 265216 | 2052562 |
| DR | 1 | 1 | 8337 | 6291 | 94427 | 3 | 11 | 99042 | 123661 | 1408820 | 1 | 1 | 2366 | 26971 | 99090 | 156923 | 1602337 |
| EP | 1 | 1 | 7185 | 5363 | 86238 | 2 | 10 | 105741 | 129339 | 1020384 | 1 | 1 | 1211 | 37952 | 126240 | 172654 | 1232862 |
| HW | 1 | 1 | 8028 | 5025 | 90652 | 1 | 4 | 31048 | 38964 | 331431 | 1 | 4 | 7589 | 44129 | 342029 | 88118 | 764112 |
| SST | 1 | 1 | 7850 | 6045 | 144748 | 3 | 16 | 177812 | 203966 | 2055904 | 1 | 1 | 1791 | 45443 | 241747 | 255454 | 2442399 |

Table 10: Output-token (OtTk) comparison between CLIMATEAGENT and a GPT-5 baseline for Task 1 in each domain. OtTk (CLIMATEAGENT) corresponds to TotO in Table 9 (total output tokens summed over Plan+Prog+Viz). GPT-5 baseline reports total output tokens for the single-pass run.

| Task | CLIMATEAGENT OtTk | GPT-5 OtTk |
|---|---|---|
| AR | 265216 | 62461 |
| DR | 156923 | 30343 |
| EP | 172654 | 35489 |
| HW | 88118 | 11358 |
| SST | 255454 | 36569 |
| Total | 938365 | 176220 |

isolate whether the observed gains come from our *climate-specific* design choices (specialized data agents, explicit persistence, artifact contracts, and targeted self-correction), or simply from using *any* multi-step agentic workflow. To address this, we include a comparison against a widely used **general-purpose multi-agent framework** built for tool-using, long-horizon tasks: **DeepAgent** implemented with LangGraph (via the `deepagents` package). DeepAgent is prompted with the *same tools* and required to satisfy the *same output/evaluation contract* as CLIMATEAGENT, enabling a controlled assessment of whether domain-specialized design is necessary beyond generic multi-agent orchestration.

### H.1 DeepAgent as a General-Purpose Multi-Agent Baseline

We use **DeepAgent** as our general-purpose multi-agent baseline. DeepAgent is an LLM-driven **supervisor** implemented as a LangGraph runnable that natively supports planning, tool use, and hierarchical delegation. In our setting, the DeepAgent supervisor can: (i) maintain explicit planning state (e.g., a TODO list via default planning tools such as `write_todos`/`read_todos`), (ii) call tools (Python functions exposed as LangChain tools), (iii) operate over a scoped filesystem backend, and (iv) delegate work to named subagents.

**Hierarchical delegation via native subagents.** At DeepAgent creation time, we define a set of named subagents, each with a description, system prompt, and a restricted subset of tools. During execution, the supervisor decides *when* to delegate by calling the native subagent tool (shown as `task` in our traces). Each subagent then runs its own tool-calling loop (within the same filesystem scope) and returns a result to the supervisor. Conceptually:

$$\text{Supervisor} \rightarrow \texttt{task}(\text{Subagent}) \rightarrow (\text{project tools}) \rightarrow \text{Artifacts}.$$

**Fairness constraints: same tools, same output contract.** To make the comparison controlled and low-risk, we enforce the same evaluator-facing artifacts and directory layout as CLIMATEAGENT: both systems write into `app/report-generate/new-approach/experiments/task_*/`

and must produce a non-empty `code_output/final_report.md`. We scope side effects with `FilesystemBackend(root_dir=task_dir)`, preventing reads/writes outside a single experiment folder. We also avoid exposing a general shell/execute tool; instead we provide a controlled `python_run` tool that executes Python under `task_dir` with a timeout.

**Project tools exposed to DeepAgent.** We expose a small, explicit tool set that mirrors the capabilities used by CLIMATEAGENT while remaining framework-agnostic:

- `init_task_dir`, `inspect_user_data`

- `context_patch` (deep-merge updates into `context.json`)

- `register_code`, `register_result` (update `context.json['codes']` and `context.json['results']`)

- `python_run` (execute a Python file or code string under `task_dir`, with a timeout)

- `validate_contract` (checks required keys and `code_output/final_report.md`)

- `get_cds_datasets` (returns curated CDS dataset identifiers)

- (optional wrappers) `run_download_cds(task_dir, subtask, attempt)` and `run_download_ecmwf(task_dir, subtask, attempt)`

These tools allow DeepAgent to manage state, create/register artifacts, run controlled code, and (optionally) reuse the same download wrappers as CLIMATEAGENT, while leaving high-level planning and delegation to the general-purpose framework.

**Named subagents.** We define four named subagents with restricted tool subsets to reflect common roles in tool-using workflows:

- **planner**: produces and updates a structured `subtasks` list in `context.json` (via `context_patch`).

- **downloader**: acquires datasets under `data/` and registers scripts/results in context.

- **analyst**: writes analysis scripts under `code/` and produces intermediate artifacts under `code\_output/`.

- **reporter**: produces `code_output/final_report.md` and verifies the output contract (via `validate_contract`).

**Observability and traces.** To support qualitative diagnosis of generic-agent behavior, the DeepAgent baseline emits structured traces: `task_*/log/deep_native.log` (runner logs), `task_*/log/deep_native_trace.jsonl` (structured events such as messages/tool calls/subagent start-end/TODO updates), and `task_*/log/deep_native_trace.txt` (a human-readable timeline). These traces enable analysis of whether errors arise from generic orchestration (e.g., planning drift, inefficient tool use, brittle path assumptions) versus domain-specific mechanisms implemented in CLIMATEAGENT.

In the next subsection, we report preliminary experiments comparing CLIMATEAGENT to DeepAgent under identical tool access and the same output contract.

## H.2 Preliminary Experiments: ClimateAgent vs. DeepAgent under the Same Tools and Output Contract

We conducted preliminary experiments comparing CLIMATEAGENT against DeepAgent under identical tool access and the same evaluator-facing output contract (Sec. H.1). Table 11 summarizes report-quality scores for six representative domains.

Table 11: Report quality comparison between CLIMATEAGENT and DeepAgent. "–" indicates `err` (DeepAgent run failed or did not produce a valid report for scoring).

| | CLIMATEAGENT | | | | | DeepAgent | | | | |
|---|---|---|---|---|---|---|---|---|---|---|
| Category | Read. | Rigor | Comp. | Vis. | Score | Read. | Rigor | Comp. | Vis. | Score |
| AR | 7 | 6 | 5 | 8 | 6.50 | 8 | 6 | 5 | 7 | 6.50 |
| DR | 8 | 9 | 7 | 8 | 8.00 | – | – | – | – | – |
| EP | 8 | 9 | 7 | 9 | 8.25 | – | – | – | – | – |
| HW | 9 | 8 | 9 | 8 | 8.50 | 9 | 9 | 8 | 7 | 8.30 |
| SST | 9 | 10 | 9 | 8 | 9.00 | 9 | 9 | 9 | 8 | 8.75 |
| TC | 9 | 8 | 9 | 8 | 8.50 | – | – | – | – | – |

**Overall outcome.** DeepAgent successfully produced scorable reports for **3/6** tasks (AR, HW, SST), while **3/6** runs failed (`err`) and did not generate a valid report for scoring (DR, EP, TC). In contrast, CLIMATEAGENT produced valid reports across all six tasks.

**Failure modes of a general-purpose framework.** The three DeepAgent failures are qualitatively consistent with known brittleness of generic multi-step agents in tool-augmented scientific workflows:

- **DR: tool invocation error.** The run failed due to incorrect tool usage (tool-call schema/arguments mismatch), preventing the workflow from progressing to subsequent steps.

- **EP: data download error.** The run failed during data acquisition, reflecting difficulty in constructing valid requests against constrained, evolving climate data APIs.

- **TC: file-path error.** The run failed at a later stage due to a path/working-directory mismatch, i.e., a brittle artifact contract between steps (an intermediate file existed but was referenced under an incorrect relative path).

These failures directly motivate two core design choices in CLIMATEAGENT: (i) **Data-Agents with dynamic API introspection** (querying the latest valid parameters / dataset metadata at runtime to reduce invalid-parameter and format-mismatch errors during download), and (ii) **Adaptive Self-Correction**, which uses bounded retries and error-conditioned regeneration to recover from runtime failures (including tool/API errors and artifact-contract issues).

**Quality comparison on successful runs.** On the three tasks where DeepAgent completed end-to-end execution (AR, HW, SST), DeepAgent scores are slightly below CLIMATEAGENT in overall report score and/or key dimensions (Table 11). For example, DeepAgent matches CLIMATEAGENT on AR report score but trails on visual quality, and it is modestly lower than CLIMATEAGENT on HW and SST overall score. While preliminary, this gap suggests that beyond generic multi-step orchestration, **domain-specific knowledge integration** (e.g., climate-aware data handling conventions, robust scientific computation patterns, and task-specific validation/formatting constraints) contributes to higher scientific rigor/completeness and more reliable end-to-end outputs.

**Takeaway.** Taken together, these preliminary tests indicate that improvements are not simply due to adopting *any* multi-step agentic workflow. A general-purpose multi-agent framework with the same tools can still fail due to tool-call brittleness, constrained data APIs, and artifact-contract/path issues, whereas CLIMATEAGENT's domain-specialized components (dynamic API introspection and adaptive self-correction) improve both completion reliability and report quality under the same output contract.

