# OpenReview forum: "ClimateAgent: Multi-Agent Orchestration for Complex Climate Data Science Workflows"
_TMLR — Accepted by TMLR_

### Review · Reviewer_H7nP · 2025-12-17

**Summary Of Contributions:**

This paper presents CLIMATEAGENT, a multi-agent framework for executing complex climate data science workflows. It uses a hierarchical architecture with specialized agents for planning, orchestration, data access, and coding to address domain-specific data heterogeneity, evolving APIs, and long-horizon execution failures. The authors also introduce CLIMATE-AGENT-BENCH-85, a benchmark of 85 expert-curated real-world climate tasks across six domains. Experiments show that CLIMATEAGENT achieves promising task completion and generates higher-quality scientific reports than GPT-5 and GitHub Copilot.

### Strengths
- The Adaptive Self-Correction strategy is a strong practical contribution. The paper clearly maps distinct failure modes (e.g., invalid API parameters, logical errors in code) to targeted recovery strategies, such as multi-candidate generation (m=8) for parameter discovery and iterative refinement for code correction. This design effectively addresses the brittleness of long-horizon scientific workflows.
- Clear Modular Architecture: The decomposition into specialized roles (Plan vs. Data vs. Coding) effectively isolates context and expertise, preventing the context drift often seen in monolithic agent workflows.
- The benchmark itself is likely to be a valuable community resource.

### Weaknesses
- Limited Baseline Comparisons: the evaluation compares CLIMATEAGENT primarily against single-model baselines (GPT-5 and GitHub Copilot). As a result, it remains unclear whether the observed gains stem from the specific architectural innovations or simply from the general advantages of a multi-agent system over a zero-shot LLM.
- Incremental Novelty: while the paper frames execution robustness as a major open challenge, many proposed components, such as browser-based parameter discovery, reflection loops, and candidate sampling, closely resemble established engineering practices in general-purpose coding agents. The work’s primary contribution appears to be thoughtful system integration and domain adaptation rather than fundamentally new AI methodologies.
- Cost: the computational and monetary costs of dynamic introspection and multi-candidate execution are not quantified, leaving questions about scalability.

**Audience:**

Yes

**Audience Explanation:**

While the paper is heavily applied, its contributions to robust agent design, benchmarking methodologies, and tool-use strategies make it relevant to a wide swath of the TMLR audience, particularly those focused on systems, evaluation, and applied ML.

**Claims And Evidence:**

Yes

**Claims Explanation:**

The claims are generally well-supported by clear and accurate evidence, though the strength of the convincing factor is slightly limited by baseline choices.

- The claim of superior performance is supported by Table 2
- The efficacy of Adaptive Self-Correction is convincingly evidenced by an ablation analysis that explicitly maps 35 baseline failure cases (e.g., Data Request Errors) to specific code fixes generated by the system.
- Qualitative evidence (Figure 2) effectively demonstrates the system's ability to maintain cross-step dependencies in complex tasks where baselines fail to produce valid visualizations.

**Requested Changes:**

- Baseline Comparison: The current evaluation compares the proposed multi-agent system against single-model baselines (GPT-5 and GitHub Copilot). To validate the claim that the specific hierarchical architecture of CLIMATEAGENT is superior, please include a comparison (or at minimum, a detailed qualitative discussion supported by preliminary tests) against a general-purpose multi-agent framework prompted with the same tools. This is necessary to determine if the performance gains stem from your specific design choices or simply from using any multi-step agentic workflow.
- Cost and Latency Analysis: Please add an analysis of the computational cost (e.g., token usage/inference cost) and wall-clock time for CLIMATEAGENT compared to the baselines. The use of multi-candidate generation (m=8) and iterative debugging implies a significant resource overhead that should be quantified.
- Clarification of Related Work: Refine the claim regarding the "gap between high-level planning and execution robustness" in the Related Work section. Please explicitly distinguish your contribution from existing general-purpose self-correction/reflexion frameworks (e.g., Reflexion, self-debugging agents) and explain why those specific mechanisms are insufficient for climate APIs without your proposed architecture.

---

> ### Author Response · Authors · 2026-02-06
>
> Thank you for the constructive feedback; we address the requested changes by adding a general-purpose multi-agent baseline comparison (Appendix H), quantifying cost/latency (Appendix G; Tables 9–10), and clarifying how our domain-specialized design differs from prior self-correction/reflexion approaches (Appendices B.5/E/H).
>
> ### W1 – Baseline Comparison
> We agree that a comparison to a general-purpose multi-agent framework with the same tools is necessary to isolate architectural gains from generic multi-step execution. We therefore add Appendix H (Comparison to a General-Purpose Multi-Agent Framework; LangGraph DeepAgent), evaluating DeepAgent under identical tool access and the same output/evaluation contract. The results show DeepAgent fails on multiple tasks due to tool-call errors, data download failures, and path/contract mismatches, whereas \sys completes the tasks under the same protocol. On tasks where DeepAgent does succeed, \sys achieves higher report quality. This supports that our gains come from domain-specialized mechanisms (e.g., API introspection, domain-specific knowledge integration and robustness-oriented recovery), not merely “using any multi-agent workflow”.
>
> ### W2 – Cost and Latency Analysis
> We agree and add a dedicated cost/latency analysis in Appendix G (Computation Cost). Table 9 reports per-domain and per-agent breakdowns (Planning/Programming/Visualization), including number of LLM invocations, input/output tokens, and end-to-end wall-clock time. Across AR/DR/EP/HW/SST, ClimateAgent uses 664,896 input tokens and 938,365 output tokens (1,603,261 total), with 8,094,272 ms (≈134.9 min) aggregated wall-clock time; the Programming agent dominates due to iterative debugging and multi-candidate generation. Table 10 further compares output tokens against a single-pass GPT-5 baseline (Task 1 per domain), showing the additional overhead associated with intermediate reasoning and retries. We include these measurements to quantify scalability trade-offs alongside robustness gains.
>
> ### W3 – Clarification of Related Work
> Our main contribution is a domain-specialized, executable architecture (ClimateAgent) and practical design insights for making autonomous end-to-end climate workflows reliable. Prior self-debugging/reflexion agents mainly target reasoning/code correction under relatively stable tool interfaces and static tasks. In climate workflows, dominant failures stem from (i) dynamic API parameter spaces requiring runtime metadata introspection, (ii) format/toolchain heterogeneity (e.g., backend/engine mismatches), and (iii) long-horizon artifact dependencies that break under brittle paths/contracts. Addressing these requires persistent external state, domain-specific data agents, and artifact-aware coordination beyond prompt-level reflection alone. We provide concrete evidence via API introspection and fallback scripting (Appendix B.5), failure cases (Appendix E), and comparisons to a general-purpose multi-agent framework under the same protocol (Appendix H).
>
> ### W4 – Incremental Novelty
> We admit that several primitives we use (e.g., candidate sampling, and reflection-style debugging) resemble established practices in general-purpose coding agents. Our contribution is a successful, concrete development of a domain-specific autonomous agent (ClimateAgent) justified by **accurate, convincing and clear empirical evidences.**
>
> Furthermore, our practice also provides design insights on how to build such systems (e.g., coordinated planning, context persistence/coordination, and self-correction). In climate workflows, previous generic agentic tools often fail due to heterogeneous data sources, strict/dynamic API parameter spaces, and fragile toolchains, and our work shows how these constraints can be addressed in a runnable end-to-end system with structured workflow decomposition, dynamic API introspection, domain-specific knowledge integration, and iterative refinement mechanisms. We believe these insights are broadly useful to researchers developing agentic systems for other scientific and data-centric domains.

---

### Review · Reviewer_RBNs · 2025-12-17

**Summary Of Contributions:**

\subsection*{Summary of Contributions}

The paper presents a structured and well-motivated approach to automating climate science workflows using large language models. Its main contributions can be summarized as follows:

\begin{itemize}
    \item \textbf{Conceptual Multi-Agent Framework.}
    The paper proposes \textit{ClimateAgent}, a conceptual multi-agent orchestration framework that decomposes complex climate science workflows into specialized roles, including planning, orchestration, data acquisition, coding, and visualization. The framework emphasizes modularity, task decomposition, and role specialization tailored to climate data analysis.

    \item \textbf{Formalization of Climate Workflows as Sequential State Transitions.}
    The authors formalize end-to-end climate analysis as a sequential transformation of a persistent workflow context, providing a clear abstraction for reasoning about planning, execution, and error recovery in long-horizon scientific workflows.

    \item \textbf{Robustness-Oriented Design Principles.}
    The work highlights three core capabilities—coordinated task planning, contextual coordination, and adaptive self-correction—and illustrates how these principles can mitigate common failure modes of LLM-based code generation in scientific settings.

    \item \textbf{Climate-Agent-Bench-85 Benchmark.}
    The paper introduces a new benchmark consisting of 85 realistic, end-to-end climate science tasks spanning six major climate domains. The benchmark is carefully curated by domain experts and reflects real-world data access, processing, and analysis challenges.

    \item \textbf{Comprehensive Empirical Evaluation.}
    Through extensive experiments, the paper demonstrates that a system following the proposed agentic design principles substantially outperforms strong GPT-5–based baselines and GitHub Copilot in terms of report quality, completeness, and robustness on the proposed benchmark.

\end{itemize}

Overall, the paper contributes a compelling architectural blueprint and evaluation framework for agent-based automation in climate science, supported by large-scale empirical evidence, though the concrete implementation details of the proposed system are not fully specified.

**Audience:**

No

**Audience Explanation:**

While the paper addresses an important application domain (climate science) and explores agent-based automation using large language models, the primary contribution is largely conceptual and application-specific rather than methodological. The work does not introduce new learning theory, model architectures, or generalizable machine learning techniques that would be of broad interest to the core TMLR audience.

Much of the paper focuses on system design choices, workflow orchestration, and domain-specific benchmarking for climate data analysis. Although these aspects may be valuable to practitioners working at the intersection of climate science and AI systems, they are less aligned with TMLR’s emphasis on foundational advances in machine learning research.

As a result, while the paper may be of interest to a specialized subset of researchers in scientific workflow automation or applied AI for climate science, it is unlikely that a substantial portion of TMLR’s general readership would find the findings directly relevant.

**Broader Impact Concerns:**

without providing a software package / or a toolbox is not possible!

**Claims And Evidence:**

No

**Claims Explanation:**

\subsection*{Major Concern: Lack of Concrete Implementation of \textit{ClimateAgent}}

While the manuscript consistently claims the introduction of an autonomous multi-agent system termed \textit{ClimateAgent}, it remains unclear where and how this system is concretely implemented beyond a conceptual design. The paper provides detailed descriptions of agent roles (Plan-Agent, Orchestrate-Agent, Data-Agent, Coding-Agent), but does not sufficiently specify whether these agents correspond to independent software components, persistent LLM instances, or logical prompt roles executed sequentially within a single control loop.

\paragraph{Implementation Ambiguity.}
The manuscript lacks a clear description of the software realization of \textit{ClimateAgent}. It is not specified whether agents are implemented as separate processes, modular classes, state machines, or simply as prompt templates. While Algorithm~1 outlines a high-level orchestration procedure, it remains abstract and does not clarify how agents are instantiated, scheduled, or managed at runtime.

\paragraph{Agent vs.\ Prompt Role Distinction.}
It is ambiguous whether each agent represents a truly independent agent with persistent memory and execution state, or whether the system consists of a single LLM assuming different roles via prompting. This distinction is critical for evaluating the novelty and validity of the claimed multi-agent architecture, as role-based prompting does not necessarily constitute a genuine multi-agent system.

\paragraph{Lack of Implementation Evidence.}
The manuscript does not provide a code repository, implementation appendix, or execution trace demonstrating the existence of a functioning \textit{ClimateAgent} system. While the benchmark results are extensive, they primarily evaluate output quality rather than the system mechanics themselves, leaving uncertainty about reproducibility and deployability.

\paragraph{Unclear Context Persistence Mechanism.}
The paper claims persistent workflow context (e.g., serialized JSON state), yet no concrete example or schema is shown. It is therefore unclear how context is stored, updated, validated, or recovered during execution, and how this mechanism differs from standard conversational memory.

\paragraph{Benchmark-System Relationship.}
Although \textit{Climate-Agent-Bench-85} is well-defined, it is not explicitly stated whether \textit{ClimateAgent} is a reusable system that can be independently applied to this benchmark, or a bespoke experimental pipeline constructed specifically for the reported evaluation. This raises concerns regarding generalizability and reuse by the community.

\paragraph{Autonomy Claims.}
The manuscript claims fully autonomous, end-to-end execution without human intervention, yet does not clarify how practical constraints such as API credentials, rate limits, tool installation, or unexpected runtime failures are handled. These omissions make it difficult to assess whether the reported autonomy is achievable in real-world settings.

\paragraph{Summary.}
Overall, \textit{ClimateAgent} currently appears more as a conceptual orchestration framework than a clearly instantiated agentic system. The paper would benefit substantially from an explicit description of the system implementation, including agent instantiation, execution flow, context management, and a minimal reproducible example or code release. Therefore I would reject the draft.

**Requested Changes:**

\subsection*{Requested Changes}

To strengthen the contribution and make the claims of an autonomous multi-agent system verifiable and reusable, the following changes are required:

\begin{itemize}
    \item \textbf{Concrete Software Implementation.}
    The authors should provide a concrete implementation of \textit{ClimateAgent} as a runnable software system or toolbox. This implementation should explicitly instantiate the proposed agents (Plan-Agent, Orchestrate-Agent, Data-Agent, Coding-Agent) as executable components rather than conceptual roles.

    \item \textbf{Use of Established Agent Frameworks.}
    Implementing the system using existing agent frameworks (e.g., LangChain, AutoAgent, CrewAI, or comparable multi-agent toolkits) would substantially improve clarity, reproducibility, and credibility. Clearly specifying how agents communicate, store context, and recover from errors is essential.

    \item \textbf{Explicit LLM Backend Specification.}
    The manuscript should clearly document which large language models are used (e.g., LLaMA-family models, GPT-based models, or other open-weight LLMs), how they are accessed, and how model choice affects system behavior. If multiple backends are supported, this should be demonstrated explicitly.

    \item \textbf{Executable Examples and Traces.}
    At least one end-to-end executable example should be provided, including the full agent interaction trace (e.g., context evolution, retries, API introspection steps). This would allow readers to verify that the system operates as claimed.

    \item \textbf{Public Code Release.}
    A public code repository with installation instructions, configuration files, and minimal reproducible scripts should be released. This is particularly important given the scale of the benchmark and the strong performance claims.

    \item \textbf{Clarification of Autonomy Assumptions.}
    The authors should clarify practical assumptions required for autonomy, such as API credentials, rate limits, tool installation, and failure handling. This clarification would help distinguish between conceptual autonomy and deployable autonomy.

\end{itemize}

Overall, providing a concrete, reproducible software implementation—ideally grounded in established agent frameworks—would significantly strengthen the paper and align its claims with verifiable system behavior.

---

> ### Author Response · Authors · 2026-02-06
>
> Thank you for the detailed feedback; to address the reproducibility/toolbox concern, we provide an anonymized repository (link: https://anonymous.4open.science/r/climate-llm-report-2B31) and add concrete implementation details, runnable commands, and end-to-end traces in Appendices B–C.
>
> ## W1 – Implementation Ambiguity
> We admit that Algorithm 1 was a little imprecise. We add Appendix B.1–B.2 to map it to the concrete runtime: a single-process Python orchestrator that schedules modular agents and executes generated scripts via subprocess with bounded retries. Agents are instantiated as executable modules (not prompt-only roles), and each run writes a complete per-task workspace for inspection. These details make the system mechanics explicit and reproducible.
>
> ## W2 – Agent vs. Prompt Role Distinction
> CLIMATEAGENT is not one LLM role-playing multiple roles via prompting. Each agent is a separately invoked component with its own I/O contract and tool procedure, coordinated by the orchestrator (Appendix B.3). State is externalized (persistent `context.json` + artifacts), not conversational memory (Appendix B.4–B.6). We also provide a `--no-context` ablation that disables passing prior summaries, making artifact dependence explicit (Appendix B.3/B.4).
>
> ## W3 – Lack of Implementation Evidence
> We add concrete, inspectable execution evidence in Appendix B.6–B.7. Each run produces `context.json`, generated code, downloaded data/metadata, logs, and final outputs, plus a structured per-call trace (`token_usage.json`) (Appendix B.6). We further provide an end-to-end executable run with retries and artifacts in Appendix C. This addresses reproducibility beyond final report quality.
>
> ## W4 – Unclear Context Persistence Mechanism
> We specify the persistence mechanism in Appendix B.4. The workflow context is an explicit serialized state (`context.json`) updated after planning and after each subtask, recording the ordered subtasks and per-step code paths/results. Failures are written into the state and trigger bounded retries; after exhaustion the run stops with an explicit failure message (Appendix B.4). Resumption is supported by continuing from a specified subtask index (Appendix B.4/B.7).
>
> ## W5 – Benchmark–System Relationship
> ClimateAgent is a reusable, climate-tailored agent system that autonomously plans tasks, routes subtasks to specialized sub-agents (data/programming/visualization), and generates a final report, rather than a bespoke pipeline for this benchmark. CLIMATE-AGENT-BENCH-85 items are executed by running the same system on benchmark prompt files, without task-specific hard-coding. Evaluation is decoupled from execution: a separate harness reads generated artifacts and scores reports without affecting system behavior (Appendix B.7). We make this interface explicit and provide runnable commands for fresh runs, resumption, and ablations (Appendix B.7).
>
> ## W6 – Autonomy Claims / Practical Assumptions
> We clarify that autonomy assumes a pre-configured environment (credentials, tools/packages, network). Within this setting, the system runs without human intervention using bounded retries, multi-candidate generation, and subprocess error capture (Appendix B.1–B.6). For strict/evolving climate APIs, we implement runtime metadata introspection and fallback script generation (Appendix B.5). We also document realistic failure modes and mitigations in Appendix E.
>
> ## W7 – TMLR Audience Interest
> We believe the work is of interest to a relatively considerable amount of TMLR audience—particularly researchers working on agentic systems, tool use, and reliable long-horizon automation—and our results suggest that domain-specific agent design can be important for reliable end-to-end automation when the domain involves heterogeneous data sources, strict API parameter spaces, and fragile toolchains. Our contribution is a successful, concrete development of a domain-specific autonomous agent (ClimateAgent) and the potentially resulting practical design insights on how to build such systems (e.g., structured workflow decomposition, dynamic API introspection, domain-specific knowledge integration, and iterative refinement mechanisms). We believe these insights are broadly useful to researchers developing agentic systems for other scientific and data-centric domains.
>
> ## W8 – Executable Examples and Traces
> We agree that a concrete executable example is essential. We add Appendix C (Executable Examples and Traces) with a complete end-to-end run, including subtask order, retries, and artifact paths. The appendix shows context evolution, generated scripts, intermediate outputs, and the final report, making execution mechanics verifiable. This provides direct evidence that ClimateAgent is a runnable system, not a conceptual description.

---

> ### Comment · Action_Editor_V9Pm · 2026-02-13
>
> Dear reviewer,
>
> Can you submit your final recommendation?
>
> Best,
> AE

---

### Review · Reviewer_EjuM · 2026-01-09

**Summary Of Contributions:**

This paper introduces CLIMATEAGENT, an autonomous multi-agent framework designed to execute end-to-end climate data science workflows, from data acquisition to analysis and report generation. The work is motivated by the observation that generic LLM-based agents struggle in climate science due to the heterogeneity of data sources, domain-specific APIs, and methodological constraints, while manual workflows remain time-consuming and do not scale. To support systematic evaluation, the authors also introduce CLIMATE-AGENT-BENCH-85, a manually curated benchmark of 85 real-world climate science tasks spanning multiple phenomena and requiring full end-to-end execution.

Strengths:
1. The paper is easy to read and well structured. The overall system architecture is clearly described, and the roles of the different agents are explained with enough details.
2. The use of specialized agents for planning, data acquisition, and coding is well justified given the complexity and heterogeneity of climate science workflows. The authors do a good job of explaining the difficulties of the climate science domain for end-to-end workflows, even for readers who are not experts in this domain.
3. The experimental section provides both quantitative results and qualitative discussion that help explain why, at a high level, the proposed multi-agent design leads to better end-to-end performance compared to generic LLM-based agents.
4. CLIMATE-AGENT-BENCH-85 is manually curated and targets end-to-end report generation rather than isolated subtasks. This fills a clear gap in the current evaluation landscape for climate science automation and represents, in my view, a major contribution of the paper.

Weaknesses:
1. While the paper provides thorough system-level analysis and discussion of performance gains, it does not include ablation studies that isolate the contributions of individual components. Additional analysis attributing improvements to specific components would strengthen the empirical claims and deepen the understanding of the system.
2. The system is evaluated exclusively on CLIMATE-AGENT-BENCH-85, which is introduced in this work. While the benchmark itself is valuable, the paper does not clearly discuss whether other climate science benchmarks or task suites exist that could be used for complementary evaluation, nor does it explicitly justify why evaluation is restricted to this single benchmark. As a result, it is difficult to assess how representative the reported performance is compared to other works in this domain.
3. The reported performance improvements over baselines such as GitHub-Copilot and a GPT-5 are clear, but their practical significance is difficult to interpret. These baselines are generic coding or agentic tools rather than climate-specific workflow systems, and the paper should make clearer why climate-focused agentic approaches cannot be used as baselines.
4. Report quality is assessed exclusively using an LLM-as-a-judge framework. Given that CLIMATE-AGENT-BENCH-85 is a central contribution and that the benchmark size is moderate, the evaluation would be significantly strengthened by including some degree of human expert assessment. As it stands, the scalar scores are somewhat difficult to interpret scientifically.
5. An analysis on system failures is missing.

**Audience:**

Yes

**Audience Explanation:**

The paper addresses a relevant problem for the TMLR community, at the intersection of LLM-based agents, scientific workflow automation, and climate science. Furthermore, this work introduces a new benchmark that can be used, after addressing the reported weaknesses, to compare different systems in end-to-end climate science tasks.

**Claims And Evidence:**

Yes

**Claims Explanation:**

The main claims of the paper are supported by clear and generally convincing evidence. The system architecture is described in sufficient detail to justify the design choices, and the experimental results on the proposed benchmark demonstrate improved end-to-end performance over strong generic baselines. The authors complement quantitative results with qualitative analyses and discussions that help explain why the proposed multi-agent approach is effective.

**Requested Changes:**

Critical changes:
- Address Weakness #4: report quality is currently assessed exclusively using an LLM-as-a-judge framework. Given that CLIMATE-AGENT-BENCH-85 is a central contribution and that the benchmark size is moderate, the evaluation would benefit from additional validation.
- Address Weakness #3: make clearer why other climate-specific workflows are not used as baselines.

Suggested changes to strengthen the work:
- Address Weakness #1: include component-level ablation or analysis.
- Address Weakness #2: either expand the evaluation to other climate-specific benchmarks to prove that the system would work, or explain more clearly why this cannot be done.
- Address Weakness #5: given the complexity of the overall system, the work would be strengthened by a dedicated analysis of failure cases, including scenarios in which the system fails to complete tasks or produces suboptimal outputs, and a discussion of the underlying causes.

---

> ### Author Response · Authors · 2026-02-06
>
> Thank you for the constructive and encouraging feedback!
>
> ### W1 – Component-level ablation or analysis
> We agree that the original submission did not sufficiently isolate the contributions of individual components. In response, we added Appendix D (Component-level Ablation Analysis) with both direct ablation (where the system remains executable) and trace-based attribution. Specifically, Appendix D.1 reports trace-grounded evidence for Coordinated Task Planning (CTP) on representative long-horizon tasks, including plan structure, artifact hand-offs, routing, and recovery traces. Appendix D.2 provides a controlled ablation for Contextual Coordination  across domains, showing consistent score drops and an end-to-end failure (“err”) when context alignment is removed. Finally, Appendix D.3 analyzes Adaptive Self-Correction via retry counts and an error taxonomy aggregated from logs, linking recovered error classes to concrete failure patterns. Together, these additions strengthen component-level attribution and clarify which design elements drive end-to-end gains.
>
> ### W2 – Evaluation scope beyond CLIMATE-AGENT-BENCH-85
> We agree that complementary evaluation would be valuable, and we will clarify this point in the revision. Our setting targets complicated real-world end-to-end executable workflows (climate data acquisition → preprocessing → analysis → visualization → report generation), where failures often arise from evolving domain APIs, strict request constraints, heterogeneous formats, and long-horizon dependencies. In contrast, the existing climate benchmarks we are aware of primarily evaluate isolated subtasks on static datasets and typically do not instantiate real-world acquisition/toolchain constraints. Because these test suites do not match the executability requirements we study, the corresponding benchmark results would not be informative to evaluate the design and implementation of CLIMATEAGENT comparing with other baselines. We therefore introduce CLIMATE-AGENT-BENCH-85 to fill this gap, and in the revision we will add a brief benchmark landscape discussion and a criteria table explaining which requirements are (not) covered by prior benchmarks.
>
> ### W3 – Why not use climate-specific workflow baselines?
> We agree that comparing only against generic tools can make practical significance harder to interpret. Based on our study, we did not find publicly available climate-specific agentic  systems that can be run under the same end-to-end execution protocol (heterogeneous CDS/ECMWF acquisition, xarray-based processing, external toolchains, and report generation) with a shared output contract. To provide a stronger agentic baseline, we conduct additional comparison and reported in Appendix H (Comparison to a General-Purpose Multi-Agent Framework; LangGraph DeepAgent). Concretely, we compare ClimateAgent to DeepAgent under identical tool access and the same output contract, isolating gains from domain-specific design choices rather than tool availability. The results show that generic multi-agent frameworks frequently fail due to schema/path/toolchain mismatches, while our domain-specialized mechanisms improve end-to-end completion. We will also make this baseline-selection rationale explicit in the revised main text.
>
> ### W4 – LLM-as-a-judge only; need additional validation
> We agree that relying exclusively on LLM-as-a-judge can make report-quality scores harder to interpret scientifically. To make the validation more reliable, we added a human expert assessment and quantified its agreement with the LLM judge in Appendix F (Human Expert vs. LLM-as-Judge Agreement). Appendix F reports mean absolute score differences between expert and LLM scores across domains, showing generally close agreement and highlighting the more challenging AR domain where discrepancies are larger. We additionally include a targeted qualitative analysis to explain the primary source of disagreement in that domain (e.g., visual evidence quality). These results provide complementary validation of the judge-based metric and clarify how to interpret the scalar scores.
>
> ### W5 – Failure-case analysis
> We agree that a dedicated failure analysis is important for an end-to-end multi-agent pipeline. To address this, we added Appendix E (Failure Cases) with four representative failures, each including log evidence, intermediate artifacts, root causes, and mitigation attempts. The cases cover external service/API failures, data/format heterogeneity (e.g., engine mismatches), runtime constraints (invalid candidates/timeouts), and brittle artifact contracts (path/CWD mismatches leading to pipeline completion failures). In addition, Appendix D.3 summarizes recovered error patterns via retry statistics and an error taxonomy from logs, complementing the case studies with an aggregate view. We will add a short pointer in the revised main text to connect our self-correction discussion to these systematic failure analyses.

---

### Decision · Action_Editor_V9Pm · 2026-03-23

**Recommendation:** Accept with minor revision

**Audience:**

Yes

**Audience Explanation:**

The paper addresses a relevant problem for the TMLR community, at the intersection of LLM-based agents, scientific workflow automation, and climate science. Furthermore, this work introduces a new benchmark that can be used, after addressing the reported weaknesses, to compare different systems in end-to-end climate science tasks.

**Claims And Evidence:**

Yes

**Claims Explanation:**

This paper presents CLIMATEAGENT, an automated multi-agent system built to handle complete climate data science projects, from gathering data to analyzing it and writing reports. The authors created this because general AI agents often have a hard time with climate science due to the wide variety of data types, specialized tools, and strict scientific rules involved. At the same time, doing this work by hand takes too long and is hard to expand. To properly test their system, the authors also introduce CLIMATE-AGENT-BENCH-85, a hand-picked set of 85 real-world climate tasks that cover different weather events and require the system to complete the entire process from start to finish.

While the paper is heavily applied, its contributions to robust agent design, benchmarking methodologies, and tool-use strategies make it relevant to a wide range of the TMLR audience, particularly those focused on systems design, agentic system evaluation, and applied ML.

The rebuttal has resolved the concerns from reviewers EjuM and H7nP. The reviewer V9Pm didn't submit his/her final recommendation. But his/her comments provide good insights. The authors provided the rebuttal, which the AE believes can mostly solved the concerns.

The AE believes that open source the project can solve the remaining concerns, which is strongly encouraged by the AE